# Long non-coding RNAs direct the SWI/SNF complex to cell type-specific enhancers

James A. Oo[1,2,3], Timothy Warwick [1,2,3], Katalin Pálfi[1], Frederike Lam [1,2,3], Francois McNicoll[4], Cristian Prieto-Garcia [5,6], Stefan Günther [7], Can Cao [1,8,9], Yinuo Zhou[1,2,3], Alexey A. Gavrilov[10], Sergey V. Razin[10,11], Alfredo Cabrera-Orefice [1,12], Ilka Wittig [1,12], Soni Savai Pullamsetti[13,14], Leo Kurian [1,2,3], Ralf Gilsbach [1,8,9], Marcel H. Schulz [2,3,15], Ivan Dikic [3,5,6], Michaela Müller-McNicoll [3,4,16], Ralf P. Brandes [1,2,3] & Matthias S. Leisegang [1,2,3] ✉

The coordination of chromatin remodeling is essential for DNA accessibility and gene expression control. The highly conserved and ubiquitously expressed SWItch/Sucrose Non-Fermentable (SWI/SNF) chromatin remodeling complex plays a central role in cell type- and context-dependent gene expression. Despite the absence of a defined DNA recognition motif, SWI/SNF binds lineage specific enhancers genome-wide where it actively maintains open chromatin state. It does so while retaining the ability to respond dynamically to cellular signals. However, the mechanisms that guide SWI/SNF to specific genomic targets have remained elusive. Here we demonstrate that *trans*-acting long non-coding RNAs (lncRNAs) direct the SWI/SNF complex to cell type-specific enhancers. SWI/SNF preferentially binds lncRNAs and these predominantly bind DNA targets in *trans*. Together they localize to enhancers, many of which are cell type-specific. Knockdown of SWI/SNF- and enhancer-bound lncRNAs causes the genome-wide redistribution of SWI/SNF away from enhancers and a concomitant differential expression of spatially connected target genes. These lncRNA-SWI/SNF-enhancer networks support an enhancer hub model of SWI/SNF genomic targeting. Our findings reveal that lncRNAs competitively recruit SWI/SNF, providing a specific and dynamic layer of control over chromatin accessibility, and reinforcing their role in mediating enhancer activity and gene expression.

The SWI/SNF chromatin remodeling complex is central in chromatin dynamics and transcription control[1]. Despite being highly evolutionarily conserved and ubiquitously expressed, SWI/SNF orchestrates cell type- and context-specific gene expression programs, cell differentiation, and lineage specification[1–5]. Not only does SWI/SNF actively maintain open chromatin states genome-wide, it also rapidly responds to cellular cues to modify genomic regions as needed[2,3,6]. It is therefore unsurprising that degradation, pharmacological inhibition and loss-of-function mutations of SWI/SNF lead to a rapid loss of chromatin accessibility, which can drive the development and progression of diseases, not least cancer[2,3,7,8].

Despite its significance, the mechanisms determining SWI/SNF complex genomic targeting in the control of cell type- and context-specific gene expression programs remain elusive. The targets of SWI/SNF themselves are DNA regulatory elements, with enhancer regions being enriched over other regions[1,9–11]. SWI/SNF-dependent regulation

of enhancers allows for the fine-tuning of highly cell type-specific genes, which are spatially connected to enhancers through 3D genome conformations[1,12]. Of note, SWI/SNF binding at enhancers and promoters is highly dynamic and changes during the cell cycle[13]. This involves a redistribution of SWI/SNF subunits from enhancers to promoters during mitosis in order to maintain critical gene transcription programs through cell division[14]. Moreover, it has recently been suggested that SWI/SNF establishes and maintains chromatin accessibility of spatial contacts at core enhancers involved in cancer[9,15]. How these processes are coordinated on a large yet specific scale is unknown.

Suggested mechanisms of SWI/SNF targeting to genomic sites include histone tail recognition[16], nucleosome acidic patch interaction[17], SWI/SNF subunit interactions[18] as well as the recruitment by pioneering transcription factors[10,11,19,20]. While all of these mechanisms likely function together, they fail to adequately address the issue of cell type-specific and genome-wide SWI/SNF targeting. Various studies including our own have explored the roles of individual long non-coding RNAs (lncRNAs) in SWI/SNF genome targeting[21–26]. LncRNAs are increasingly understood to control gene expression, often through interactions with DNA-binding proteins and chromatin itself[27–30]. The emerging role of lncRNAs in SWI/SNF function highlights a unique layer of regulation whereby lncRNAs might serve as precision guides that target SWI/SNF to specific genomic sites to control chromatin structure and gene expression. The function of lncRNAs in the recognition and establishment of enhancer-promoter loops is also becoming evident, for example, through the base-pairing of complementary sequences in enhancer RNAs and promoter-derived non-coding RNAs[31,32]. The interaction of lncRNAs with proteins and DNA is dictated in part by lncRNA sequence specificity, structure and modification profile. Moreover, lncRNA expression is often highly species-, cell type- and context-specific; as is the case for enhancers and their regulation[27,33]. As such, lncRNAs are well positioned to mediate the specific targeting of SWI/SNF to genome-wide enhancers.

While there are indications, to our knowledge no study has mapped SWI/SNF chromatin occupancy, functional SWI/SNF-RNA interactions, and SWI/SNF-RNA-DNA interactions in a primary cell type. Here, we performed an unbiased screen of SWI/SNF-RNA-DNA interactions in endothelial cells (HUVEC) followed by the functional analysis of those interactions. On this basis, the study demonstrates how lncRNAs guide SWI/SNF to its target genomic sites, revealing a fundamental layer of specificity in chromatin remodeling and transcriptional regulation.

## Results

### Trans-acting lncRNAs are enriched with SWI/SNF at cell type-specific enhancers

Given the unknown role of RNA in SWI/SNF targeting to specific genome-wide sites, we first performed Red-C (RNA ends on DNA capture) and RedChIP (Red-C with chromatin immunoprecipitation (ChIP))[34,35]. Red-C mapped global RNA-DNA interactions while RedChIP, targeting the SWI/SNF core ATPase BRG1, enabled the identification of BRG1-enriched RNA-DNA interactions (Fig. 1a). Compared to Red-C, BRG1 RedChIP exhibited a narrower yet more precise genomic and transcriptomic coverage (Fig. 1b, c and Supplementary Fig. 1a–c), reflecting its specificity for BRG1-bound sites. BRG1 RedChIP also recovered a greater number of higher-confidence RNA-DNA interactions, with normalized interaction frequencies greater than 10 (Fig. 1d), and identified more genomic bins with higher summed interaction frequencies (Fig. 1e). These results demonstrate an enrichment of RNA-DNA interactions at BRG1-bound genomic regions across multiple loci. This enrichment raises the question of whether these interactions are relevant to SWI/SNF targeting and function.

While RedChIP effectively maps proteins at RNA-DNA interaction sites, its resolution was limited to 5 kb to enable sufficient mapping of RNA binding sites to genomic features. To identify the precise DNA binding sites of BRG1, we used BRG1 CUT&RUN data[36] (Fig. 1a). Within these BRG1 CUT&RUN binding sites, there was an enrichment of Red-C RNA-DNA interactions compared to shuffled BRG1 CUT&RUN binding sites (Supplementary Fig. 1d and e), even before RedChIP was performed. This already demonstrates the co-localization of BRG1 and RNA at the same DNA sites, and was further enhanced by BRG1 RedChIP (Fig. 1f and Supplementary Fig. 1d and e). Integration of BRG1 CUT&RUN and BRG1 RedChIP data revealed that BRG1 DNA binding sites are predominantly associated with *trans*-acting RNAs (Fig. 1g). Additional analysis of those DNA binding sites revealed an enrichment of the *trans*-acting RNAs at intergenic regions of DNA, where enhancers are harbored (Fig. 1h). Conversely, intronic BRG1 binding sites were enriched for *cis*-acting RNAs, with BRG1 promoter peaks displaying a balanced interaction across all RNA classes. Notably, in contrast to enhancers, some promoter BRG1 interaction sites lack RNA association, suggesting lower RNA dependence at promoters compared to enhancers (Fig. 1h). Given the unique pattern of RedChIP intergenic RNA-DNA enrichment, we next analyzed primary cell ChIP-sequencing datasets[37,38] to identify HUVEC-specific DNA features. Remarkably, the SWI/SNF-associated *trans*-RNAs were indeed localized to enhancers (Fig. 1i) and many of these were even specific to HUVEC when compared to six other cell types of equivalent epigenomic quality (Fig. 1j). The majority of HUVEC enhancers were classed as strong enhancers while they were largely repressed or inactive in the other cell types (Fig. 1j). The unique binding patterns of BRG1-associated lncRNAs are exemplified by *MALAT1, MIR100HG* and *PVT1* RedChIP browser traces. Their RNA-DNA binding sites overlap with BRG1 CUT&RUN peaks (Fig. 1k). For example, *PVT1* and *MIR100HG* bind two separate sites where a BRG1 peak is centered. These findings reveal a selective binding of the SWI/SNF complex at RNA-DNA interaction sites, with a notable enrichment of *trans*-acting RNAs and SWI/SNF at HUVEC-specific enhancers compared to promoters. This pattern reflects an RNA-mediated targeting mechanism that directs SWI/SNF to cell type-specific enhancers.

### SWI/SNF preferentially binds trans-acting lncRNAs

To corroborate the RedChIP findings and identify high-confidence SWI/SNF-RNA interactions for further stratification, we performed individual-nucleotide resolution crosslinking and immunoprecipitation (iCLIP)[39,40]. BRG1 iCLIP was performed to accurately evaluate the RNA interactions. As a whole, BRG1 iCLIP coverage increased with RNA expression (Fig. 2a). However, normalization of BRG1 iCLIP coverage to both RNA-seq coverage and RNA length revealed lncRNAs as the only positively enriched RNA class (Fig. 2b). 3859 high-confidence RNAs were identified after setting a threshold on iCLIP coverage (counts per million > 5) (Fig. 2c). There were also significantly more observed BRG1 lncRNA binding sites than expected (based on their proportion in RNA-seq expression data), while there was no difference between expected and observed mRNA BRG1 binding sites (Fig. 2d). This indicates that SWI/SNF preferentially binds lncRNAs, warranting further investigation into the role of lncRNAs in SWI/SNF genomic targeting. To determine the functional relevance of lncRNAs in SWI/SNF recruitment, we stratified the top lncRNA binders based on the following criteria: presence of the lncRNA in both Red-C and BRG1 RedChIP datasets, presence of BRG1 iCLIP binding sites and expression across HUVEC replicates in the RNA-seq dataset. LncRNAs were then ranked compared to a mean scaled value for all lncRNAs for their degree of iCLIP coverage (FPKM), their expression level (FPKM), normalized enrichment score (iCLIP normalized to expression and RNA length) and number of iCLIP binding sites (Fig. 2e). The top selected 16 lncRNA candidates were higher expressed, enriched and contained more binding sites compared to the mean of all lncRNAs (Fig. 2f).

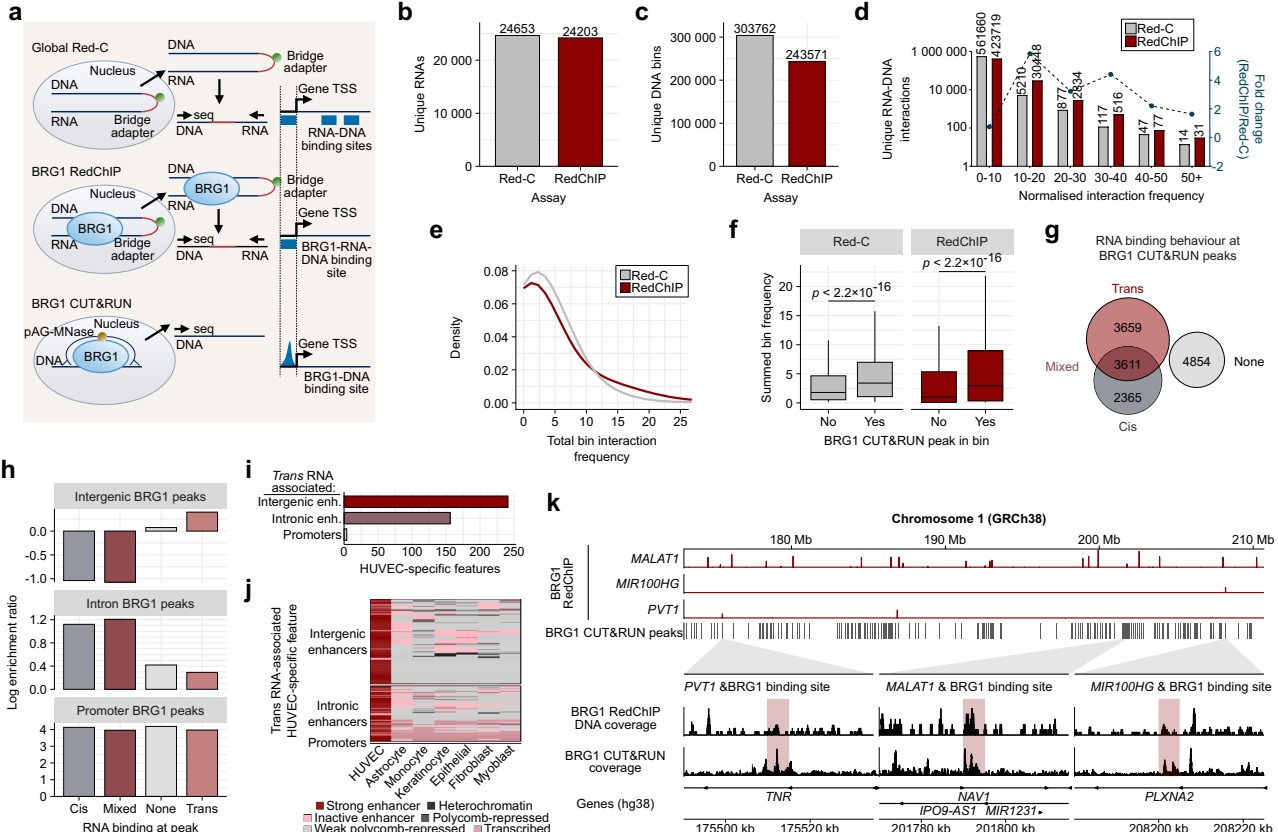

**Fig. 1 | Trans-acting lncRNAs are enriched with SWI/SNF at cell type-specific enhancers. a** Schematic representation of the Red-C, RedChIP and CUT&RUN experimental workflow. HUVEC underwent crosslinking, followed by bridging of DNA and RNA ends with an oligonucleotide-adapter. BRG1 was pulled down, followed by a biotin pulldown of the RNA-DNA-bridge adapter interactions. Red-C served as a control for the global RNA-DNA background. BRG1 CUT&RUN provided high-confidence BRG1 DNA binding sites. TSS (Transcription Start Site), seq (sequencing), pAG-Mnase (protein A and G Micrococcal Nuclease). **b, c** Quantitative assessment of unique RNAs (**b**) and DNAs (**c**) involved in the interactions identified by Red-C and BRG1 RedChIP. **d** Unique RNA-DNA interactions identified by Red-C and BRG1 RedChIP, and corresponding fold-change, in interaction frequency bins. **e** Comparison of the number of unique RNA-DNA interactions density across increasing interaction frequency bins. **f** Frequency of both Red-C and RedChIP RNA-DNA interactions within BRG1 CUT&RUN peaks. Box plots are defined as follows: "Red-C False" minima: 0.1791, maxima: 10.7481, center: 1.7913, lower bound (25th percentile): 0.5374, upper bound (75th percentile): 4.6575; "Red-C True" minima: 0.1791, maxima: 15.7638, center: 3.4036, lower bound (25th percentile): 1.0748,

upper bound (75th percentile): 6.9862; "RedChIP False" minima: 0.0837, maxima: 13.2273, center: 1.0046, lower bound (25th percentile): 0.0837, upper bound (75th percentile): 5.3579; "RedChIP True" minima: 0.0837, maxima: 21.8502, center: 2.9301, lower bound (25th percentile): 0.3349, upper bound (75th percentile): 8.9577. Whiskers extend from lower bound to minima and from upper bound to maxima. A Wilcoxon signed-rank test was performed within each condition (n = 1, Red-C and RedChIP). **g** Classification of BRG1 CUT&RUN peaks based on the absence (None) or presence of *trans*-acting or *cis*-acting RNAs. **h** Enrichment of RNA classes (*trans*, *cis*, mixed or none) across different DNA elements (intergenic, intronic and promoter) bound by BRG1. **i** Absolute number of *trans*-acting RNA-DNA interaction sites within HUVEC-specific features (intergenic enhancers, intronic enhancers and promoters). **j** Heatmap of HUVEC-specific features where *trans*-acting RNAs bind, with the classification of each feature in different cell types. **k** Visualization of selected BRG1-associated lncRNAs (*PVT1*, *MALAT1* and *MIR100HG*) from RedChIP and their proximity to BRG1 CUT&RUN peaks. The red window superimposed onto the BRG1 CUT&RUN trace represents a 5 kb RNA binding site from RedChIP. Source data are provided as a Source Data file.

Comparison of these iCLIP lncRNA candidates with the BRG1 RedChIP dataset revealed a predominance of *trans*-acting behaviors, aligning with the observation that SWI/SNF-bound RNAs are mostly *trans*-acting (Fig. 2g). Example strand-specific iCLIP browser traces are provided for lncRNAs *PVT1* and *LINC00607* that demonstrate the binding of BRG1 (Fig. 2h). These results highlight a preference of the SWI/SNF complex for *trans*-acting lncRNA interactions and underscores the potential regulatory significance of *trans*-acting lncRNAs in SWI/SNF genomic targeting.

### Knockdown of SWI/SNF-bound lncRNAs causes a global genomic redistribution of SWI/SNF

To explore the role of *trans*-acting lncRNAs on SWI/SNF localization at gene regulatory elements, we utilized an siRNA library to knock down eight SWI/SNF-bound lncRNAs (Supplementary Fig. 2a), and subsequently performed BRG1 CUT&RUN. Each lncRNA knockdown resulted in unique BRG1 binding profiles, with both a loss and gain of

BRG1 CUT&RUN peaks genome-wide (Fig. 3a). The differential BRG1 peaks varied between the lncRNAs studied, which clustered into two main groups based on the magnitude of their effect on BRG1 binding. The concomitant loss and gain of BRG1 binding sites was unexpected, given our original hypothesis that BRG1-bound lncRNAs primarily recruit BRG1 to genomic targets. This observation led us to consider the possibility that SWI/SNF recruitment might be competitively regulated by multiple lncRNAs, or that the complex might default to other DNA sites upon specific lncRNA depletion. The gain of BRG1 peaks after lncRNA knockdown could also suggest a local deterrence of SWI/SNF by lncRNAs. At mixed *cis/trans* binding sites, *trans* RNA depletion is sufficient to attenuate BRG1 binding at that site, as demonstrated at *SMG7* and *PCDH9* intronic regions after depletion of *LINC00607* and *PVT1* respectively (Supplementary Fig. 2b). By leveraging ChromHMM[37], BRG1 CUT&RUN data was integrated with existing histone modification ChIP-seq datasets, allowing for the identification of chromatin states and DNA element

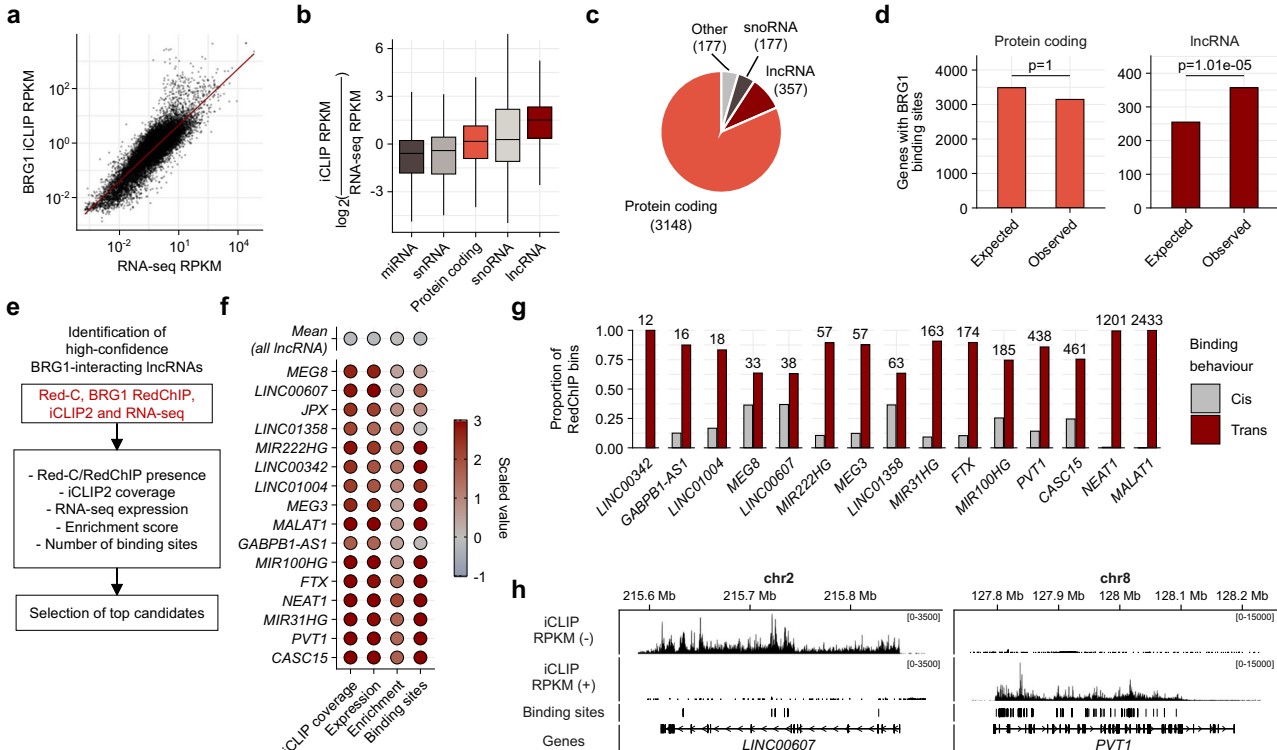

**Fig. 2 | SWI/SNF preferentially binds *trans*-acting lncRNAs. a** Correlation between BRG1 iCLIP coverage and RNA-seq coverage, normalized to RNA length. Reads per kilobase million (RPKM). **b** Analysis of RNA classes based on normalized iCLIP coverage. Log₂(iCLIP coverage/RNA-seq coverage) values are plotted to compare RNA classes (lncRNAs, mRNAs, snoRNAs, snRNAs, and miRNAs). Reads per kilobase million (RPKM). Box plots are defined as follows: "miRNA" minima: -4.8725, maxima: 3.2751, center: -0.5896, lower bound (25th percentile): -1.8241, upper bound (75th percentile): 0.2226; "snRNA" minima: -4.4778, maxima: 3.1295, center: -0.4113, lower bound (25th percentile): -1.8856, upper bound (75th percentile): 0.4326; "Protein coding" minima: -3.9667, maxima: 4.2052, center: 0.1668, lower bound (25th percentile): −0.9103, upper bound (75th percentile): 1.1378; "snoRNA" minima: −4.9670, maxima: 6.9218, center: 0.2799, lower bound (25th percentile): −1.0903, upper bound (75th percentile): 2.1851; "lncRNA" minima: −2.5777, maxima: 5.2470, center: 1.5063, lower bound (25th percentile): 0.3676, upper bound (75th percentile): 2.3317. Whiskers extend from lower bound to minima and from upper bound to maxima. **c** Proportion of RNA classes enriched in

BRG1 iCLIP (counts per million > 5). **d** Statistical analysis comparing the expected (based on relative basal expression) versus observed frequency of BRG1 binding sites within lncRNAs and mRNAs. Fisher's exact test was performed within each condition, $n = 4$ technical replicates. **e** Stratification to identify the top bound lncRNAs. Initial screen for lncRNA presence in Red-C, BRG1 RedChIP, BRG1 iCLIP and RNA-seq. LncRNAs were then ranked compared to a mean scaled value for all lncRNAs for their degree of iCLIP coverage (RPKM), their expression level (RPKM), normalized enrichment score (iCLIP normalized to expression and RNA length) and number of iCLIP binding sites. **f** Scaled value plots for the top 16 lncRNA candidates based on selection criteria compared to the mean scaled value for all identified lncRNAs. **g** Comparative analysis of *trans*- and *cis*-acting lncRNAs identified in BRG1 RedChIP and iCLIP datasets. **h** Visualization of selected lncRNAs (*PVT1* and *LINC00607*) and their BRG1 iCLIP strand-specific binding profiles. Called iCLIP binding sites are displayed. Reads per kilobase million (RPKM). Source data are provided as a Source Data file.

types. Untransfected control and negative control siRNA conditions both demonstrated an enrichment of BRG1 at enhancers under basal conditions (Fig. 3b), consistent with previous findings that SWI/SNF preferentially binds enhancers[1,9,10]. Remarkably, five lncRNA knockdowns (*PVT1*, *MIR31HG*, *LINC00607*, *JPX* and *MIR100HG*) led to a substantial number of differential BRG1 peaks and also a redistribution of BRG1 enrichment from enhancers to promoters (Fig. 3b), suggesting that these lncRNAs play a pivotal role in directing SWI/SNF to enhancer sites.

This is exemplified by example browser traces where *MIR100HG* (Fig. 3c), *JPX* (Fig. 3d) and *PVT1* (Fig. 3e) knockdowns all show an attenuation of BRG1 binding at enhancers. There are also enhancers with more BRG1 binding after lncRNA knockdown, such as that observed for *LINC00342* knockdown (Fig. 3f). Importantly, the loss of BRG1 binding at specific sites could be rescued by the overexpression of a *JPX* or *MIR100HG* transcript following siRNA-mediated depletion of the respective lncRNA (Fig. 3g and h). This highlights the specificity for each lncRNA and DNA binding site and the diverse regulatory mechanisms at play depending on the sites and molecules involved. These findings demonstrate the regulatory influence of lncRNAs on

SWI/SNF DNA binding, prompting further investigation into enhancer dynamics and gene expression.

## LncRNAs retain BRG1 at functional SWI/SNF-dependent enhancers
To investigate SWI/SNF dependency at specific enhancers and their functional relevance, we treated cells with the PROTAC AU-15330 to degrade BRG1, effectively disabling SWI/SNF complexes (Supplementary Fig. 3a)[9]. Subsequent ATAC-seq revealed a significant change in chromatin accessibility: within 30 minutes of PROTAC treatment, 25,280 ATAC peaks were lost and 2972 gained, with peak loss doubling by 60 minutes and stabilizing by 240 minutes (Fig. 4a). The rapid collapse of open chromatin regions reflects the active maintenance of these sites by SWI/SNF, highlighting gene regulatory elements dependent on SWI/SNF activity. Exemplary siRNA-mediated knockdown of *MIR100HG* and CRISPR-Cas9-mediated knockout of *LINC00607* also led to a collapse of certain chromatin regions that were sensitive to the PROTAC, reflecting a similar dependence of chromatin accessibility on lncRNAs (Supplementary Fig. 3b).

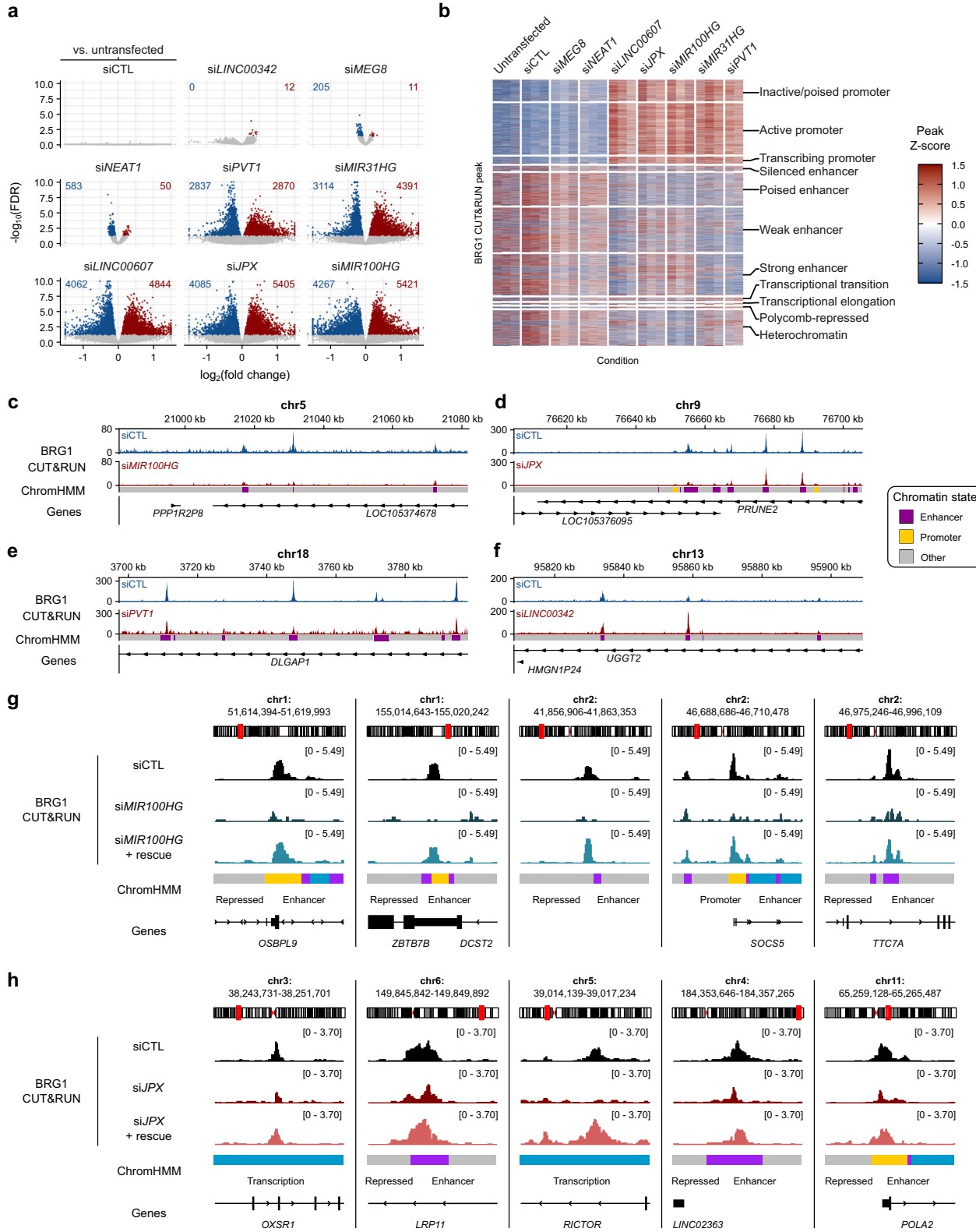

Grouping the gene regulatory elements into enhancers and promoters revealed that most lost BRG1 sites were at enhancers, with 6000 regions closing within 30 min and increasing to 16,000 by 60 min (Fig. 4b). Conversely, only a minor fraction of promoter regions closed, even after 240 min of PROTAC treatment (Fig. 4b). The majority of BRG1-bound promoters maintained accessibility and BRG1 binding. Of the promoters that lost BRG1, their open chromatin state was mostly maintained, indicating a potential compensatory mechanism. Such compensation and recovery after SWI/SNF inhibition can be promoter autonomous, not reliant on nearby enhancers[41]. Collectively, these findings reveal that SWI/SNF-bound promoters are able to maintain open chromatin and baseline expression even when SWI/SNF binding is disrupted. This contrasts with enhancers, which are more reliant on SWI/SNF for their activity and lack similar

**Fig. 3 | Knockdown of SWI/SNF-bound lncRNAs causes a global genomic redistribution of SWI/SNF. a** BRG1 CUT&RUN lost (blue) and gained (red) peaks following lncRNA knockdowns. Negative control siRNA (siCTL) compared to untransfected control (top left). siRNA lncRNA knockdowns compared to siCTL. **b** ChromHMM chromatin state analysis of BRG1 binding patterns after lncRNA knockdown. **c–f** Visualization of specific lncRNA knockdowns illustrating BRG1 binding changes. Example traces for *MIR100HG* (**c**), *JPX* (**d**), and *PVT1* (**e**) knockdowns highlight reduced BRG1 binding at enhancers (indicated by purple

rectangles), while increased BRG1 binding at enhancers is observed in *LINC00342* knockdown (**f**). **g** Browser traces of BRG1 CUT&RUN after siRNA-mediated knockdown of *MIR100HG* and knockdown followed by overexpression rescue with a *MIR100HG* transcript. 5 different genomic regions annotated with ChromHMM are shown. siRNA control (siCTL). **h** Browser traces of BRG1 CUT&RUN after siRNA-mediated knockdown of *JPX* and knockdown followed by overexpression rescue with a *JPX* transcript. 5 different genomic regions annotated with ChromHMM are shown. siRNA control (siCTL). Source data are provided as a Source Data file.

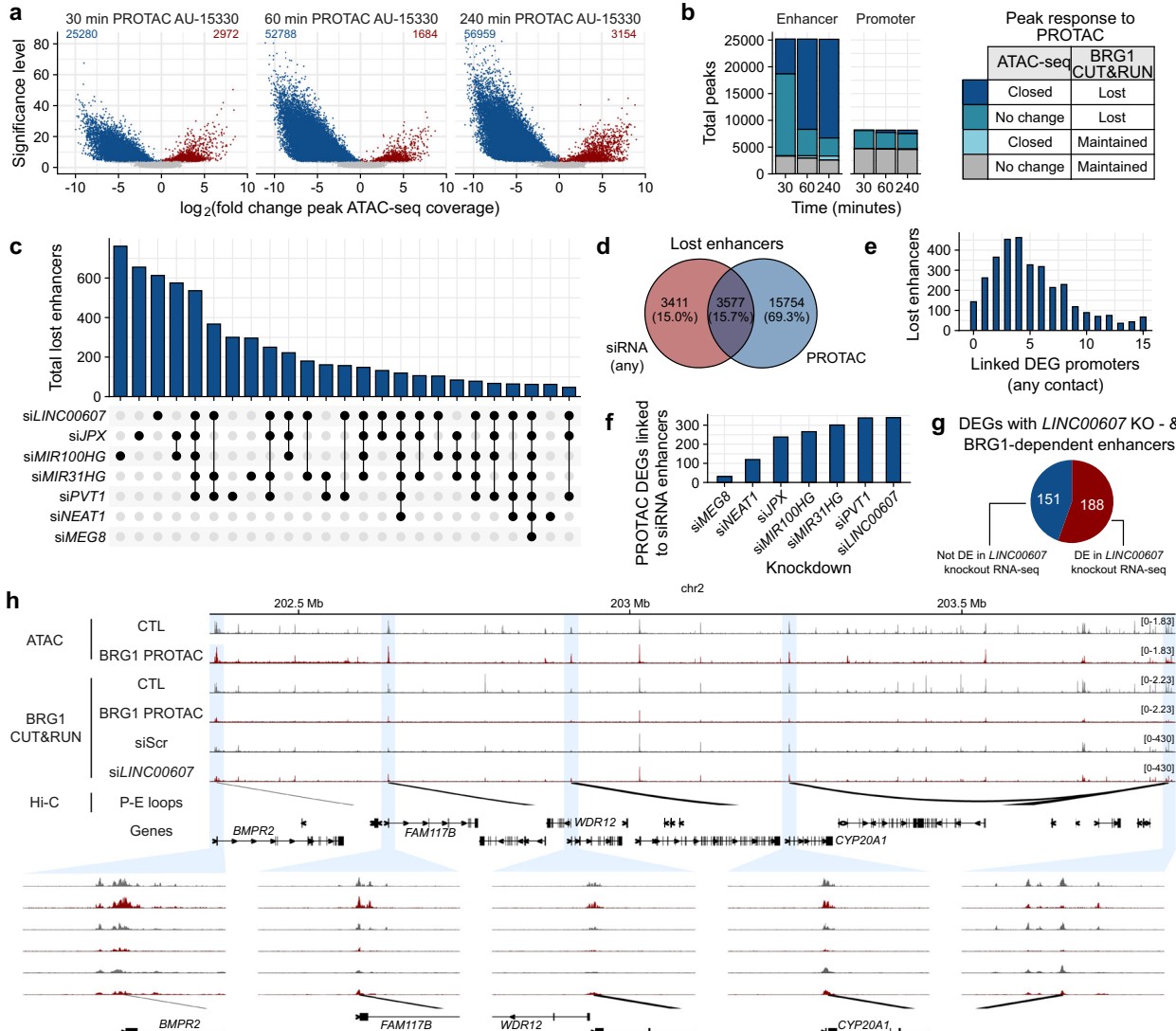

**Fig. 4 | LncRNAs retain BRG1 at functional SWI/SNF-dependent enhancers. a** Volcano plots of differential ATAC-seq peaks after treatment with a PROTAC against BRG1 and BRM ATPases. 30, 60 and 240 minutes of PROTAC treatment are displayed. **b** BRG1 binding (CUT&RUN) and corresponding chromatin accessibility (ATAC-seq) at enhancers and promoters after PROTAC treatment. **c** Upset plot of total lost enhancers for each lncRNA knockdown and their overlap. **d** Enhancers lost with any lncRNA siRNA, PROTAC and both. **e** Number of DEG (Differentially Expressed Gene) promoters associated with enhancers perturbed by both siRNA of

lncRNAs and PROTAC. **f** Number of DEGs after PROTAC that loop to enhancers lost after siRNA knockdown of individual lncRNAs. **g** Number of BRG1-dependent enhancers lost after *LINC00607* CRISPR knockout (KO), overlapped with the differential expression (DE) of the connected genes after *LINC00607* siRNA. **h** Browser visualization of an enhancer (right-most) sensitive to PROTAC (lost BRG1 CUT&RUN binding and diminished ATAC peak) and siRNA against *LINC00607*. P-E loops (Promoter-Enhancer loops). Source data are provided as a Source Data file.

compensatory mechanisms, highlighting their role in fine-tuning gene expression.

Exploring SWI/SNF-sensitive enhancers further, we next checked for those where BRG1 binding was lost after lncRNA knockdown. For example, knockdowns of *LINC00607*, *JPX*, and *MIR100HG* each led to over 600 enhancer sites losing BRG1 binding, with some enhancer losses shared across knockdowns, indicating sites where multiple

lncRNAs may function (Fig. 4c). 15.7% of the enhancer losses in response to the AU-15330 PROTAC could be attributed to the knockdown of any of the seven lncRNAs (Fig. 4d). It is tempting to speculate that the other 350 lncRNAs bound to SWI/SNF address the remaining PROTAC-lost enhancers.

Hi-C analysis was conducted to map genome-wide enhancer-promoter interactions, identifying enhancers whose loss impacts gene

regulation. Enhancers lost after PROTAC treatment and any of the lncRNA knockdowns were connected via Hi-C to gene promoters, directly linking them to PROTAC-dependent differentially expressed genes (DEGs). Most lost enhancers were associated with at least one DEG promoter (Fig. 4e), with 245 lost enhancers connected to a single DEG promoter, and the majority of lost enhancers connected to three or four promoters, supporting an enhancer hub model[12]. The number of connected PROTAC-sensitive DEGs were also a function of the number of lost enhancers after individual lncRNA knockdowns. Enhancers lost after *MEG8* knockdown for example, connected to 45 DEGs, while *PVT1* and *LINC00607* lost enhancers connected to 350 DEGs each (Fig. 4f), highlighting the specificity of lncRNA effects on SWI/SNF targeting and gene expression. RNA-seq after *LINC00607* knockdown identified 188 DEGs connected to enhancers lost following both PROTAC application and *LINC00607* siRNA (Fig. 4g). Example browser traces demonstrate one enhancer with a diminished ATAC peak after PROTAC and associated loss of BRG1 binding after PROTAC and *LINC00607* siRNA. Hi-C revealed 3D chromatin connections between the enhancer and the four gene promoters, all of which were differentially expressed after PROTAC and *LINC00607* siRNA, without significant changes in promoter BRG1 and ATAC profiles (Fig. 4h). Our data illustrate a lncRNA-regulated mechanism in chromatin dynamics, elucidating the precise nature of SWI/SNF targeting to regulatory elements genome-wide.

## Discussion

SWI/SNF is essential for chromatin remodeling and thereby gene expression control, genome stability and chromatin architecture[1–5]. However, exactly how the SWI/SNF complex is targeted to genomic sites in a cell type- and context-specific manner remains unclear. Various mechanisms of SWI/SNF recruitment have been proposed, but none fully address the high degree of specificity required for precise and contextual genome-wide chromatin dynamics. Our study demonstrates the specific targeting of SWI/SNF to gene regulatory elements by *trans*-acting lncRNAs.

It is well-established that enhancers express RNAs in the cells in which they are active, and these enhancer-derived RNAs (eRNAs) are required for proper enhancer function in a cell type-specific manner[33,42]. eRNAs are often bidirectional and correlate with enhancer activation and target gene expression[43]. Importantly, they can mediate enhancer-promoter looping, regulate chromatin accessibility, and facilitate the recruitment of transcription factors, chromatin remodelers, and other regulatory proteins to enhancers[44]. Recent research has demonstrated that eRNAs recruit SWI/SNF in *cis* to enhancers linked to cell lineage priming[18]. Our findings add to this growing body of evidence by showing that *trans*-acting lncRNAs associate with cell type-specific enhancers and play a critical role in directing the SWI/SNF complex to these regulatory regions.

In this study we performed an unbiased screen for SWI/SNF-interacting RNAs and their DNA targets. Our results uncover a layer of SWI/SNF regulation in chromatin remodeling, and underscore the complex interplay between SWI/SNF, RNA, and chromatin architecture. We demonstrated that SWI/SNF preferentially binds *trans*-acting lncRNAs at enhancers and is then lost after knockdown of the lncRNA. SWI/SNF is mostly maintained at promoters after lncRNA knockdown. Interestingly, these are the DNA elements that are least associated with RNA. The dynamic SWI/SNF binding patterns at enhancers and promoters might indicate three-dimensional chromatin interactions at these sites, especially since SWI/SNF can maintain such interactions[9,15]. Our findings suggest that lncRNAs may enable SWI/SNF maintenance of three-dimensional chromatin interactions, favoring a model of enhancer hubs in which individual enhancers connect to more than one promoter. This allows for regulatory information to be transmitted to multiple genes, facilitating coordinated gene

expression changes[12]. By demonstrating that *trans*-acting lncRNAs target SWI/SNF to specific genomic locations, we reveal a mechanism by which enhancers can exert their regulatory influence, providing texture to the role of lncRNAs in enhancer function. Our observations support the model that lncRNAs serve as critical intermediaries between enhancers and transcriptional machinery, directing SWI/SNF to enhancers and thereby modulating chromatin interactions and gene expression.

Recent studies have addressed the authenticity of methods to measure protein-RNA interactions[45,46] and DNA-RNA interactions[47]. Pulldown experiments can be unspecific and tend to produce false positives. We also considered this and opted for multiple methods to stratify high-confidence interactions. BRG1 RedChIP identified RNA-DNA-BRG1 interactions, BRG1 iCLIP identified RNA-BRG1 interactions and BRG1 CUT&RUN identified DNA-BRG1 interactions. Red-C and RedChIP revealed that RNAs tended to be enriched at their sites of transcription, serving as a proof of principle for the method. Furthermore, knockdown of individual BRG1-bound RNAs altered site-specific binding of BRG1; sites identified in both RedChIP and CUT&RUN. The use of a PROTAC revealed open chromatin sites that were most sensitive to SWI/SNF depletion and to lncRNA depletion. Sensitive sites were at enhancers, where both SWI/SNF and RNAs are known to bind[1,9,10,31]. These were linked by Hi-C to the promoters of genes differentially expressed after lncRNA depletion and PROTAC treatment. Taken together, this unbiased approach using different methods allowed for the identification of unique RNA-DNA-SWI/SNF interactions and revealed the specific targeting mechanisms of individual lncRNAs.

The SWI/SNF complex is essential in the maintenance of cellular homeostasis and the functional transcriptome under changing cellular contexts. Accordingly, its dysregulation triggers SWI/SNF deficiency-associated diseases, such as cancer. SWI/SNF mutations are found in 20% of all human malignancies and approach a mutation frequency of p53. Studies have shown that SWI/SNF can be targeted therapeutically in cancer but PROTACs and chemical inhibitors impact global SWI/SNF and could lead to many side-effects[9]. Knowledge of the lncRNAs responsible for SWI/SNF recruitment at specfic sites offers a tremendous potential for more tailored, mutation-specific therapeutics.

Taken together, we have uncovered a specific mechanism of SWI/SNF genome targeting by lncRNAs. Our model explains the specificity of gene expression control by SWI/SNF, reinforces the role of lncRNAs in mediating enhancer activity, and advocates for the potential of lncRNAs as specific targets for individual cancer therapy.

## Methods

### Cell culture and stimulation experiments

Pooled human umbilical vein endothelial cells (HUVEC) were purchased from PromoCell (C-12203, Lot No. 405Z013, 408Z014, 416Z042, Heidelberg, Germany) and cultured at 37 °C with 5% $CO_2$ in a humidified incubator. The HUVEC batches originate from umbilical cord/ umbilical vein of caucasians (405Z013: 2 males, 1 female; 408Z014: 2 males, 1 female; 416Z042: 2 males, 2 females). Gelatin-coated dishes (356009, Corning Incorporated, USA) were used to culture the cells.

Endothelial growth medium kit enhanced (EGM) (PB-C-MH-100-2199, PeloBiotech, Germany) without hydrocortisone supplemented with growth factors (EGF, bFGF, IGF, VEGF, Heparin, L-Glutamin), 8% fetal calf serum (FCS) (S0113, Biochrom, Germany), penicillin (50 U/ml) and streptomycin (50 µg/ml) (15140-122, Gibco/Lifetechnologies, USA) was used to culture HUVEC. For each experiment, at least three different batches of HUVEC from passage 3 were used.

Cells were treated at 70% confluence with 1 µM PROTAC AU-15330 (Medchemexpress, HY-145388) or DMSO in full growth medium (EGM) for 60 min.

## RNA isolation, reverse transcription and RT-qPCR

Total RNA isolation was performed with the RNA Mini Kit (Bio&Sell) according to the manufacturer's protocol and reverse transcription was performed with the SuperScript III Reverse Transcriptase (Thermo Fisher) using a combination of oligo(dT)23 and random hexamer primers (Sigma). The resulting cDNA was amplified in an AriaMX cycler (Agilent) with the ITaq Universal SYBR Green Supermix and ROX as reference dye (Bio-Rad, 1725125). Relative expression of human target genes was normalized to GAPDH. Expression levels were analyzed by the delta-delta Ct method with the AriaMX qPCR software (Agilent Aria 1.7). The following primers were used: BRG1 forward 5′-TCG CCA AGA TCC GTT GGA AG-3′ and reverse 5′-GCC ACA TAG TGC GTG TTG AG-3′, LINC00342 forward 5′-GCA GGC AGA CAA GAT TAG AG-3′ and reverse 5′-GGT AGT CCA AGC AGT CAT AG-3′, MEG8 forward 5′-AGA AGA CCA GCC TTC CAG AC-3′ and reverse 5′-CGC AGG ATA CTG AGG TGT TG-3′, NEAT1 forward 5′-CGA GGT CAG GAG TTC AAT AC-3′ and reverse 5′-CCG GGT TCA GTG ATC TTC TC-3′, PVT1 forward 5′-TGC CCA TGC CAT AGA TCC TG-3′ and reverse 5′-TGG GAG CCC GTT ATT CTG TC-3′, MIR31HG forward 5′-AAA TGC AGG CTC ACC ACA TC-3′ and reverse 5′-TCT AGG AGC AAG GAC AAA GG-3′, LINC00607 forward 5′-GGA TGC AGC AGA AGA GGA TG-3′ and reverse 5′-GAC AGG ACT GGC AGT AAT CG-3′, JPX forward 5′-AGG CGT CCG AAG TAT GAG TC-3′ and reverse 5′-TTA GGC GAT CAG CGA GAA AG-3′, MIR100HG forward 5′-TTG GAG TGT GGC AGA GTA AG-3′ and reverse 5′-ATT TAG GAA GGG CAG ACC AG-3′, GAPDH forward 5′-TGC ACC ACC AAC TGC TTA GC-3′ and reverse 5′-GGC ATG GAC TGT GGT CAT GAG-3′.

## Knockdown with siRNAs

For the knockdown of lncRNAs with small interfering RNA (siRNA), HUVEC were seeded at 15,000 cells/cm$^2$ one day before transfection. Cells were transfected using GeneTrans II according to the instructions provided by MoBiTec (Göttingen, Germany). A custom Silencer™ Select siRNA library (Thermo Fisher Scientific, Cat. No. 4392426) was used for individual knockdowns of lncRNA targets (siRNA Assay IDs n549468 (*LINC00342*), n266141 (*MEG8*), n509902 (*NEAT1*), s225363 (*PVT1*), n266043 (*MIR31HG*), s56342 (*LINC00607*), n504762 (*JPX*) and n269144 (*MIR100HG*)), with transfection performed using 25 nM siRNA. Silencer™ Select Negative Control No. 1 siRNA (Thermo Fisher Scientific, Cat. No. 4390844) served as a control. Cell medium was changed to EGM after 4 h and again the next day. All siRNA experiments were performed for 48 h.

## CRISPR/Cas9 of LINC00607

*LINC00607* KO HUVECs originated from Boos et al. [36]. Briefly, guide RNAs (gRNA) targeting *LINC00607* were designed with CRISPOR algorithm (http://crispor.tefor.net/)[48]. A dual gRNA LentiCRISPRv2 approach (one gRNA targeting a region downstream of the TSS of *LINC00607*, the other gRNA targeting a region upstream of the TSS of *LINC00607*) was used to facilitate the KO of *LINC00607*. LentiCRISPRv2 was a gift from Feng Zhang (Addgene plasmid #52961)[49]. The gRNA sequences were the following: gRNA1, 5′-CAT GTG CCC CCT TTG TTG AA-3′ and gRNA2, 5′-CAG TGT GTC ATG TTA TCT TG-3′. Lentivirus was produced in Lenti-X 293 T cells (#632180, Takara) using Polyethylenamine (#408727, Sigma-Aldrich), psPAX2 and pVSVG (pMD2.G). psPAX2 and pMD2.G were a gift from Didier Trono (Addgene plasmid #12260; Addgene plasmid #12259). LentiCRISPRv2-produced virus was transduced in HUVECs of passage 1 using polybrene (MerckMillipore, # TR-1003-G). Selection was performed with puromycin (1 μg/mL) and hygromycin (100 μg/ mL) for 6 days prior to further experiments.

## Overexpression of plasmids

For rescue experiments, plasmid overexpression (full-length pcDNA3.1 + *JPX* (NR_024582.1) or pcDNA3.1 + *MIR100HG* (isoform NR_137194.1 missing the last 1035nt) was performed 24 h after siRNA knockdown in HUVEC. The Neon electroporation system (Invitrogen)

was used according to the manufacturer's instructions. The overexpression was performed for 24 h. Full length pcDNA3.1 + *JPX* and pcDNA3.1 + *MIR100HG* were obtained from Biomatik (Canada).

## Protein isolation and Western analysis

For whole cell lysis, HUVEC were washed in Hanks solution (Applichem) and lysed with RIPA buffer (1x TBS, 1 % Desoxycholat, 1 % Triton, 0.1 % SDS, 2 mM Orthovanadat, 10 nM Okadaic Acid, protein-inhibitor mix, 40 μg/ml Phenylmethylsulfonylfluorid). After centrifugation (10 min, 16,000 × *g*), protein concentrations of the supernatant were determined by Bradford assay and the extract boiled in Laemmli buffer. Equal amounts of protein were separated with SDS-PAGE. Gels were blotted onto a nitrocellulose membrane, which was blocked afterward in Rotiblock (Carl Roth). After the application of the first antibody, an infrared-fluorescent-dye-conjugated secondary antibody (Licor) was used. Signals were detected with an infrared-based laser scanning detection system (Odyssey Classic, Licor) and Licor Image Studio (Ver 5.2) was used for analysis. Anti-BRG1 (A303-877A, Bethyl) and anti-β-actin (A1978, Sigma Aldrich) antibodies were used for immunoblotting at 1:1000 and 1:2000 dilutions respectively.

## iCLIP2

iCLIP experiments were performed using the iCLIP2 protocol[50] with minor modifications. For each replicate, cells were grown near confluence on two 150 mm culture dishes, washed with ice-cold PBS, irradiated with 300 mJ/cm$^2$ UV light at 254 nm (CL-1000, UVP), harvested by scraping and centrifugation and stored at -80 °C until lysis. Following lysis and partial digestion with RNase I (Thermo Fisher Scientific, AM2294), immunoprecipitation of BRG1 was performed using 6 μg of the anti-BRG1 (ab110641, Abcam) antibody coupled to Dynabeads™ Protein G (Thermo Fisher Scientific, 10002D). Co-purified, crosslinked RNA fragments were dephosphorylated at their 3′ ends using T4 Polynucleotide Kinase (New England Biolabs, M0201S) and ligated to a pre-adenylated 3′ adapter (L3-App). To visualize protein-RNA complexes, RNA fragments crosslinked to BRG1 were labeled at their 5′ ends using T4 Polynucleotide Kinase and γ-$^{32}$P-ATP (Hartmann Analytic). Samples were run on a Nu-PAGE 4–12% Bis-Tris Protein Gel (Thermo Fisher Scientific, NP0335BOX), transferred to a 0.45 μm nitrocellulose membrane (GE Healthcare Life Science, 10600002) and visualized using a Phosphorimager. Regions of interest were cut from the nitrocellulose membrane (above 180 kDa), and RNA was released from the membrane using Proteinase K (Roche, 03115828001). RNA was purified using neutral phenol/chloroform/isoamylalcohol (Ambion, AM9722) followed by chloroform (Serva, 39554.02) extraction, and reverse transcribed using SuperScript III™ (Life Technologies, 18080-044). cDNA was cleaned up using MyONE Silane beads (Life Technologies, 37002D) followed by ligation of a second adapter containing a bipartite (5-nt + 4-nt) unique molecular identifier (UMI) as well as a 6-nt experimental barcode[50]. iCLIP2 libraries were pre-amplified with 6 PCR cycles using short primers (P5Solexa_short and P3Solexa_short) and then size-selected using the ProNex Size-Selective Purification System (Promega, NG2001) in a 1:2.95 (v/v) sample:bead ratio to eliminate products originating from short cDNAs or primer dimers. The size-selected library was amplified for 10 cycles using P5Solexa and P3Solexa primers, and primers were removed using the ProNex Size-Selective Purification System (Promega, NG2001) in a 1:2.4 (v/v) sample:bead ratio. Purified iCLIP2 libraries were sequenced on a NextSeq 500 System (Illumina) using a NextSeq® 500/550 High Output Kit v2 as 92-nt single-end reads, yielding between 20 and 26 million reads.

## iCLIP2 analysis

Basic quality controls were performed using FastQC (version 0.11.8)[51]. The FASTX-Toolkit (version 0.0.14)[52] and seqtk (version 1.3)[53] were used to filter reads based on sequencing qualities (Phred score) in the

barcode and UMI regions. Reads were de-multiplexed according to the sample barcode on positions 6 to 11 of the reads using Flexbar (version 3.4.0, using non-default parameter --barcode-keep)[54]. Flexbar was also used to trim UMI and barcode regions as well as adapter sequences from read ends requiring a minimal overlap of 1 nt of read and adapter. UMIs were added to the read names and reads shorter than 15 nt were removed from further analysis. The downstream analysis was done as described previously[55] with an additional step to remove reads directly mapped to the chromosome ends using Samtools (version 1.9)[56] and bedtools (version 2.27.1)[57]. Those reads do not have an upstream position and, thus, no crosslink position can be extracted. Genome assembly and annotation of GENCODE (release 31)[58] were used during mapping with STAR (version 2.7.3a)[59].

Processed reads from five replicates were merged prior to peak calling with PureCLIP (version 1.3.1)[60] using a minimum transition probability of 1%. Significant crosslink sites (1 nt) were filtered by their PureCLIP score, removing the lowest 2% of crosslink sites. The remaining sites were merged into 7-nt wide binding sites using the R/Bioconductor package BindingSiteFinder (version 1.0.0), filtering for sites with at least 3 positions covered by crosslink events. Only reproducible binding sites were considered for further analysis, which had to be supported by four out of five replicates. Binding sites were overlapped with gene and transcript annotations obtained from GENCODE (release 29).

iCLIP coverage per gene was computed using bedtools (v2.27.1) coverage with GENCODE[58] annotated genes as input regions. The same procedure was performed for RNA-sequencing data in order to compare iCLIP and RNA-sequencing coverages. Gene biotype annotations were taken from the GENCODE v43 gene annotations to facilitate comparisons between lncRNAs and protein-coding RNAs. The enrichment in number of iCLIP binding sites per biotype was computed using a Fisher's exact test, where the expected number of binding sites was established by computing the proportions of gene biotypes which were expressed in the RNA-sequencing dataset (counts per million > 5).

## Red-C and RedChIP

Red-C and RedChIP were performed as previously described[35]. Briefly, $5 \times 10^7$ HUVEC were crosslinked with formaldehyde and lysed for nuclear extraction. RNA that was not crosslinked to DNA was washed away with nuclease-free water followed by the addition of SDS. Nuclei were pelleted and DNA integrity tested by loading DNA on an agarose gel after incubation with the N1aIII restriction enzyme (NEB, R0125L). After DNA restriction, RNA 3′ ends were dephosphorylated using T4 PNK (NEB, M0201). DNA ends were then blunted using T4 DNA polymerase (NEB, M0203L) and Klenow (NEB, M0210M) and then A-tailed using Klenow (exo-) (NEB, M0212M). RNA 3′ OH ends were ligated with the 5′ rApp ends of the bridge adapter (IDT, /rApp/TCC TAG CAC CAT CAA TGC GAT AGG CAA CGC TCC GAC T* and /Phos/GTC GGA GCG TTG CC/T-Biotin/ATCG) and unligated bridge adapter was washed away with multiple wash steps before ligation of the bridge adapter to DNA. Nuclei were lysed, sonicated and centrifuged and the supernatant passed through Amicon 30 K Ultra-0.5 mL Centrifugal Filters (Merck Millipore, UFC503008) by centrifugation. 5% of the concentrated supernatant was used as input (Red-C) and the remaining supernatant was divided into aliquots and made up to 1 mL with RIPA buffer (50 mM Tris pH 7.5, 150 mM NaCl, 2 mM EDTA, 0.5% SDC, 0.1% SDS, 1% NP-40, 1× protease inhibitors, 100 U SUPERase.In RNase inhibitor (Invitrogen, AM2696)). 4 μL anti-BRG1 antibody (Abcam, ab110641) was added to the supernatant aliquots overnight. Washed protein A/G beads were added to the IP reactions for 6 h. Proteinase K digestion was performed and RNA-DNA chimeras precipitated from IP and input fractions before purification with AMPure XP beads (Beckman Colter, A63882). RNA-DNA chimeras were digested with MmeI (NEB, R0637L) in a reaction containing a double-stranded oligonucleotide with an MmeI site (IDT, CTG TCC GTT CCG ACT ACC CTC CCG

AC and GTC GGG AGG GTA GTC GGA ACG GAC AG). Biotinylated bridge adapter and RNA-DNA chimeras were then pulled down using streptavidin beads (Thermo Fisher, 65002) and a switch template oligonucleotide (IDT, iCiGiC GTG ACT GGA GTT CAG ACG TGT GCT CTT CCG ATC TrGrG rG**) added to permit reverse transcription (SMART technology). Illumina adapters (AGA TCG GAA GAG CGT CGT GTA GGG AAA GAG TGT AGA TCT CGG TGG TCG CCG TAT CAT T and AAT GAT ACG GCG ACC ACC GAG ATC TAC ACT CTT TCC CTA CAC GAC GCT CTT CCG ATC TNN***) were ligated and DNA-cDNA chimeras were then amplified with the KAPA HiFi kit (Roche, Cat. No. 07958838001). PCR products were enriched for fragments > 200 bp and sequenced on a NextSeq2000 instrument (Illumina) using v2 chemistry, resulting in 90 million reads for the Red-C sample and 145 million reads for the BRG1 RedChIP sample, with a 2x100bp paired-end setup.

## RedChIP analysis

Red-C and RedChIP reads were deduplicated using FastUniq (v1.1)[61] and subsequently trimmed for quality using Trimmomatic (v0.39)[62] with the flag –phred33 and setting SLIDINGWINDOW:5:26 MINLEN:0. Reads with the appropriate linker oligonucleotide AGTCG-GAGCGTTGCCTATCGCATTGATGGTGCTAGGA were identified using SeqKit (v2.7.0)[63] grep to select reads with either the forward or reverse bridging oligonucleotide, permitting up to four mismatches. Valid 3′ RNA reads were also identified using seqkit grep and searching for either the pattern "^GGG" or "CCC$". DNA and 5′ RNA sequences were split from reads containing the bridging oligonucleotide sequence using cutadapt (v2.8)[64] with either –g or –a set to the sequence of the bridging oligonucleotide. Additionally, the –m flag was set to 14, and the –O to 37, with –e set to 0.11 to permit the mismatches allowed in the previous step. DNA reads, 5′ RNA reads and 3′ RNA reads were each aligned separately to the GRCh38 genome assembly using STAR (2.7.10)[59]. In the case of the DNA, the parameters –alignIntronMax 1 --alignMatesGapMax 1 --outFilterScoreMinOverLread 0 --outFilterMatchNminOverLread 0 --outFilterMatchNmin 0 were all set, whereas for alignment of the RNA reads, --alignIntronMax 20000 --alignMatesGapMax 20000 were both used. For all of the alignments, uniquely mapping reads were selected using samtools (v1.10)[56] view with –q set to 255. DNA alignments were mapped onto 5 kb genomic bins which were computed using bedtools (v2.27.1)[57] makewindows with GRCh38 chromosome sizes and –w 5000. RNA alignments were mapped to GRCh38 gene regions which were taken from GENCODE v43 gene annotations[65]. Only interactions where both RNA alignments were uniquely mapped to the same gene were kept for analysis. RNA-bin interaction counts were then computed, and could be compared between Red-C and BRG1 RedChIP. Interaction counts per sample were normalized based on the total number of valid interacting reads obtained per sample. Trans interactions were defined as those where an RNA was detected as interacting with DNA loci more than 5 Mb away from its gene locus. DNA bins were classified as containing a BRG1 CUT&RUN peak if there was any overlap between the bin and a called BRG1 CUT&RUN peak. Annotation and enrichment of genomic features (intergenic, intronic and promoter regions) was performed using Homer (v4.1.1) annotatePeaks[66].

## RNA-sequencing

900 ng of total RNA isolated from HUVEC was used as input for SMARTer Stranded Total RNA Sample Prep Kit–HI Mammalian (Takara Bio). Sequencing was performed on the NextSeq500 instrument (Illumina) using v2 chemistry, resulting in an average of 34 million reads per library with 1 × 75bp single end setup. The resulting raw reads were assessed for quality, adapter content and duplication rates with FastQC[67]. Trimmomatic version 0.39 was employed to trim reads after a quality drop below a mean of Q20 in a window of 10 nucleotides[62]. Only reads between 30 and 150 nucleotides were cleared for further analyzes. Trimmed and filtered reads were aligned versus the Ensembl

human genome version hg38 (release 99) using STAR 2.7.3a with the parameter "--outFilterMismatchNoverLmax 0.1" to increase the maximum ratio of mismatches to mapped length to 10 %[59]. The number of reads aligning to genes was counted with featureCounts 1.6.5 tool from the Subread package[68]. Only reads mapping at least partially inside exons were admitted and aggregated per gene. Reads overlapping multiple genes or aligning to multiple regions were excluded. Differentially expressed genes were identified using DESeq2 version 1.26.0[69]. Only genes with a minimum fold change of ± 1.5 (log2 ± 0.59), a maximum Benjamini-Hochberg corrected $p$-value of 0.05, and a minimum combined mean of 5 reads were deemed to be significantly differentially expressed. The Ensemble annotation was enriched with UniProt data (release 06.06.2014) based on Ensembl gene identifiers (Activities at the Universal Protein Resource (UniProt)[70]).

### Assay for Transposase-Accessible Chromatin using sequencing (ATAC-seq)

50,000 HUVEC were used for ATAC[71] library preparation using Illumina Tagment DNA Enzyme and Buffer Kit (Illumina). The cell pellet was resuspended in 50 µl of the lysis/transposition reaction mix (25 µl TD-Buffer, 2.5 µl Nextera Tn5 Transposase, 0.5 µl 10 % NP-40 and 32 µl H$_2$O) and incubated at 37 °C for 30 min followed by immediate purification of DNA fragments with the MinElute PCR Purification Kit (Qiagen). ATAC-seq including amplification of Library and Indexing was performed as described elsewhere[71]. Libraries were mixed in equimolar ratios and sequenced on NextSeq500 platform using V2 chemistry. Reads were aligned to the GRCh38 genome assembly using Bowtie2 (v2.4.5)[72] with the parameters–local–very-sensitive. Duplicates and multi-mapping reads were removed using samtools (v1.10)[56] rmdup and samtools view. Genomic ATAC-seq seq coverage was computed using bamCoverage (3.5.3)[73] with the parameters –normalizeUsing RPKM and –effectiveGenomeSize 2864785220. Peaks were called using MACS3 (v3.0.0)[74] with the parameters –nomodel and–format BAMPE. Differential peaks were also called using MACS3 by comparing read pileups between conditions using macs3 bdgdiff, normalized with the requisite total sequencing depths per condition. Peaks with a log likelihood greater than 3.84 were considered as differential.

### CUT & RUN

BRG1 Cleavage Under Targets & Release Using Nuclease (CUT&RUN)[75], was performed as described in the EpiCypher CUT&RUN Protocol v2.0, but with minor modifications for the cell type and antibody used. Briefly, 500,000 HUVEC were washed with CUT&RUN wash buffer (20 mM HEPES pH 7.9, 150 mM NaCl, 500 nM spermidine, 1X Roche Protein Inhibitor Cocktail) at RT. Cells were resuspended in wash buffer and 10 µl BioMag®Plus Concanavalin A (ConA) beads (Polysciences, 86057-3) were added for 10 min at RT. Beads were separated on a magnetic rack and washed once before being resuspended in 100 µl antibody buffer (wash buffer, 0.25% Digitonin and 2 mM EDTA) and 1 µl BRG1 antibody (Abcam, ab110641). Beads were incubated with the antibody overnight with gentle shaking at 4 °C. The next day, beads were washed twice with 200 µl 0.25% Digitonin wash buffer and resuspended in Digitonin wash buffer containing 2 µl CUTANA™ pAG-MNase (15-1016, EpiCypher, 15-1016) and incubated on ice for 30 min. Samples were washed twice and then resuspended in 100 µl Digitonin wash buffer containing 2 µl CaCl$_2$ at a final concentration of 100 mM and incubated for 2 h at 4 °C with gentle shaking. 33 µl of 2X "stop solution" (340 mM NaCl, 20 mM EDTA, 4 mM EGTA, 0.25 % Digitonin, 100 µg/ml RNase A, 50 µg/ml Glycoblue) was added to the beads and incubated at 37 °C for 10 min. Samples were placed on a magnetic rack and the supernatant removed and kept for DNA purification. 5X volume of binding buffer (20 mM HEPES pH 7.9, 20 mM KCl, 1 mM CaCl$_2$, 1 mM MnCl$_2$) was added to the samples and the pH adjusted with sodium acetate before being transferred to a purification column

(ActiveMotif, 58002) and centrifuged at $11,000 \times g$ for 30 sec. The column was then washed with 750 µl wash buffer and dried by centrifugation for 2 min. DNA was eluted with 25 µl elution buffer and the DNA concentration measured with a Qubit 3.0 Fluorometer (Life Technologies).

### Library preparation and sequencing of CUT&RUN samples

DNA libraries were prepared according to the manufacturer's protocol (NEBNext® Ultra II, NEB) with some minor adjustments for CUT&RUN samples. Briefly, samples were brought to 50 µl with 0.1X TE buffer and DNA end preparation performed as instructed but with incubation at 20 °C for 20 min and then 58 °C for 45 min. Adapter ligation was performed with a 1:10 dilution of the adapter (NEB, E6440S). For DNA purification, 0.9X Volume AMPure XP beads (Beckman Colter, A63881) was added to the samples and incubated for 5 min at RT. Beads were washed twice with 200 µl 80 % ethanol and DNA eluted with 17 µl 0.1X TE buffer for 2 min at RT. PCR amplification of the eluted DNA was performed as described in the manufacturer's protocol but with the addition of 2.5 µl Evagreen (20X) for visualization of the amplification curves on an AriaMx Real-time PCR system (Agilent Aria 1.7). The denaturation and annealing/extension steps of the PCR amplification were performed for around 12 cycles and stopped before the curves plateaued. A cleanup of the PCR reaction was performed twice with 1.1X Ampure beads and eluted each time in 33 µl 0.1X TE buffer. DNA concentrations were measured with a Qubit (Thermo Fisher) and size distributions measured on a Bioanalyzer (Agilent).

Sequencing was performed on the NextSeq1000/2000 (Illumina). Resulting reads were aligned to the GRCh38 genome assembly using Bowtie2 (v2.4.5)[72] with the parameters –local and –very-sensitive. Mates were fixed with samtools (v1.10)[56] fixmate with the parameter –m set, and duplicate alignments were subsequently removed using samtools markdup with the –r flag set. Reads aligning to blacklisted regions as defined by ENCODE were removed using bedtools (2.27.1)[57] intersect with the –v flag set. Resulting BAM files were indexed using samtools index. Peaks were called using MACS3 (v3.0.0)[74] with the parameters –scale-to-small –f BAM and –g hs. Differential BRG1 CUT&RUN peaks were called using the R package DiffBind (v3.10.1)[76]. Peak counts per replicate were normalized to the proportion of total reads falling within called peaks. Peak counts were then compared between conditions, and peaks with a false discovery rate-adjusted $p$-value less than 0.05 were considered as being significantly different between conditions. Peaks were annotated with genomic contexts by using ChromHMM (v1.24)[77] to compute chromatin states in HUVECs using ChIP-sequencing data from ENCODE[38], which were annotated for functionality based on published literature. CUT&RUN genomic coverage was computed using bamCoverage (v3.5.3)[73]. Normalized genomic coverage was accomplished by setting the parameter –scale-factor to the appropriate scale based on the fraction of reads in peaks detected by DiffBind.

### Hi-C

Putative enhancer-promoter interactions in HUVEC were computed using STARE (v1.0)[78] from Hi-C contacts provided by Prof. Dr. Ralf Gilsbach and Can Cao. The regions provided as potential regulatory elements were consensus peaks from ATAC-sequencing in either control or BRG1 PROTAC-treated HUVEC. The GENCODE v43 annotation[58] was used to establish gene and therefore promoter positions. Hi-C was analyzed at a resolution of 1 kb and putative enhancer-promoter interactions were classified as those where the adapted activity-by-contact (ABC) score was greater than 0.02. The ABC model integrates chromatin accessibility and histone modification data with Hi-C to calculate the liklihood of a given site being an enhancer that regulates a target gene promoter[79].

## Statistics

Unless otherwise indicated, data are given as means ± standard error of mean (SEM). Calculations were performed with GraphPad Prism 10.1.2. This was also used to test for normal distribution and similarity of variance. Individual statistics of unpaired samples by unpaired t-test. $P$ values of $< 0.05$ was considered as significant. Unless otherwise indicated, $n$ indicates the number of individual experiments.

## Reporting summary

Further information on research design is available in the Nature Portfolio Reporting Summary linked to this article.

## Data availability

All NGS datasets have been deposited and are available at NCBI GEO with the accession number GSE262070 at the following URL: https://www.ncbi.nlm.nih.gov/geo/query/acc.cgi?acc=GSE262070. ATAC-seq of LINC00607 KO and NTC was already published by Boos et al. [36] and is publicly available at NCBI GEO with the accession number GSE199878. Source data for this paper are available with the https://doi.org/10.6084/m9.figshare.27918267.v1.

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

## Acknowledgements

We thank Mitchell Guttman, Mario Blanco and Jimmy Guo for advice on our protein-RNA interaction experiments. We also thank Judit Izquierdo-Ponce for technical assistance and Anke Busch and IMB Bioinformatics Core Facility for processing the iCLIP2 data. Support by the IMB Genomics Core Facility is gratefully acknowledged. This work was supported by the Goethe University Frankfurt am Main, the DFG excellence cluster Cardio-Pulmonary Institute (CPI) EXS2026 (Projektnummer 390649896) and the DFG Transregio TRR267 (Project-ID 403584255 - TRR 267; TP A03 (M.M.-M.), TP A04 (M.S.L.), TP A06 (R.P.B.), TP Z02 (I.W.) and TP Z03 (M.H.S.)). The project was also supported by the German Center for Cardiovascular Research (DZHK, Standortproject Rhine Main and trinational project ReGenLnc, Projektnummer 81×2200165) as well as the Dr. Rolf Schwiete-Stiftung. C.C. was supported by the China Scholarship Council (202008410206). L.K. is supported by the European Research Council (ERC) Consolidator Grant (TRANSCEND, project number: GA 101043645). The IMB Genomics Core Facility and its NextSeq 500 were funded by the Deutsche Forschungsgemeinschaft [DFG, German Research Foundation] – 329045328.

## Author contributions

J.A.O., R.G., R.P.B., and M.S.L. designed the experiments. J.A.O., K.P., F.L., F.M., C.C., Y.Z., S.G. and M.S.L. performed the experiments. J.A.O., T.W., S.G., R.P.B., and M.S.L. analyzed the data. J.A.O., T.W., S.G., R.G., and M.S. performed bioinformatics. C.P.-G., A.A.G., S.V.R., A.C.-O., I.W., S.S.P., L.K., I.D. and M.M.-M. helped with research design and advice. J.A.O., R.P.B., and M.S.L. wrote the manuscript. All authors approved the manuscript.

## Funding

## Competing interests

The authors declare no competing interests.

## Additional information

[1]Goethe University Frankfurt, Institute for Cardiovascular Physiology, Frankfurt, Germany. [2]German Center of Cardiovascular Research (DZHK), Partner site Rhein/Main, Frankfurt, Germany. [3]Cardio-Pulmonary Institute (CPI), Goethe University Frankfurt, Frankfurt, Germany. [4]Goethe University Frankfurt, Institute for Molecular Biosciences, Frankfurt, Germany. [5]Goethe University Frankfurt, Institute of Biochemistry II, Faculty of Medicine, Frankfurt, Germany. [6]Goethe University Frankfurt, Buchmann Institute for Molecular Life Sciences, Frankfurt, Germany. [7]Max Planck Institute for Heart and Lung Research, Bad Nauheim, Germany. [8]Institute of Experimental Cardiology, Heidelberg University Hospital, Heidelberg, Germany. [9]German Center of Cardiovascular Research (DZHK), Partner site Heidelberg/Mannheim, Heidelberg, Germany. [10]Institute of Gene Biology, Russian Academy of Sciences, Moscow, Russia. [11]Faculty of Biology, Lomonosov Moscow State University, Moscow, Russia. [12]Goethe University Frankfurt, Functional Proteomics Center, Frankfurt, Germany. [13]Department of Internal Medicine, Justus Liebig University, Giessen, Germany. [14]Cardio-Pulmonary Institute (CPI), University of Giessen, Giessen, Germany. [15]Goethe University Frankfurt, Institute for Computational Genomic Medicine, Frankfurt, Germany. [16]Max Planck Institute for Biophysics, Frankfurt, Germany. ✉e-mail: Leisegang@vrc.uni-frankfurt.de

