## [Peer Review file · Nature Communications]

Long non-coding RNAs direct the SWI/SNF complex to cell type-specific enhancers

Corresponding Author: Dr Matthias Leisegang

Version 0:

Reviewer comments:

Reviewer #1

(Remarks to the Author)

The manuscript by Oo and colleagues investigates the concept of RNA mediating the binding of SWI/SNF complexes to its genomic targets. To address this, the authors perform a large number of NGS experiments to map RNA-DNA, protein-DNA and protein-RNA interactions focusing on the key BRG1 subunit of SWI/SNF complexes. The data is, for the most part, presented clearly and the analyses and figures are informative. The majority of conclusions are carefully drawn, and the implications of these findings come to build on the emerging concept of RNA-guided and/or RNA-assisted binding of TFs to chromatin. I find this contribution to be potentially important and of interest to a broad readership like that of Nat Communications. At the same time, I am listing below a number of points that, in my opinion, need attention in order to either strengthen or further clarify the findings of the manuscript. I hope that the authors find them useful while revising their work.

- The argument in the first paragraph that RedChIP produces more enrichment than Red-C, therefore "demonstrat[ing] a genome-wide enrichment of RNA-DNA interactions with the SWI/SNF complex" is partially flawed. Any immune-selecting method produces by definition a different structure of library for sequencing and, also by definition, a library enriched for the target of the antibody (assuming the antibody is specific and the target complex accessible, which seems to be the case for BRG1). Therefore, this line of argumentation is not correct and the fold-change calculated in Fig. 1d should not be interpreted in such a way. I would argue that RedChIP for BRG1 shows local enrichment for RNA-DNA contacts at BRG1-bound sites and nothing more in comparison to genome-wide Red-C.
- It would be also informative if the main Fig. 1 included zoomed-in snapshots of Red-C/RedChIP signal distributions from the genome browser, as well as average plots along BRG1 peaks and control non-bound regions. The statement on "limited resolution" of the Red methods would also require an explanation of which resolution is actually being used here and how this was chosen. This is important for interpreting most panels in Fig. 1 where comparisons are made.
- Regarding the composition of RNA-bound positions (presumably at low resolution bins, judging from the scale in Fig. 1k), how many different RNAs can be found bound to any given bin? How does this variety change genome-wide and according to the genomic character (genic v non-genic, HMM composition, histone mark, transcriptional activity, etc)? Do the authors believe that RNA combinations are responsible for inducing binding or rather single RNAs?
- The statement that "BRG1 DNA binding sites are predominantly associated with trans-acting RNAs" is somewhat misleading, as the numbers in Fig. 1g actually show a <18% difference in total numbers, with >3500 sites having both cis and trans RNAs mapped to them. And this also brings about another question: at these shared sites, do these cis and trans RNAs coexist? If so, how do the authors imagine them working? Given the strong enrichments of trans RNAs in intergenic regions and the enrichment of cis RNAs at intronic/intragenic regions, which positions have the most trans/cis RNA co-association? Is there any experiment that can be done to dissect this? For example, silencing of the underlying gene with dCas9-KRAB and assessment of how loss of the cis (but perhaps not of the trans RNA) affects BRG1 binding locally?
- Given that RIP-seq is a rather noisy experiment, and that the authors have already performed iCLIP, I think that all mention of RIP can be removed from text and figures as superfluous and of lower confidence. The scheme in Fig. 2e does not justify its use.
- The iCLIP profiles provided in Fig. 2h are rather broad and look a bit unspecific. There are three things to be clarified here.

First, that there is little convergence between RIP and iCLIP (which is another reason why RIP should be removed, in my opinion). Second, the peak calling for PVT1 looks rather unspecific compared to that for LINC00607. Third, what is the motif consensus outcome after using iCLIP peaks to deduce specificity in BRG1 binding? I think it not an unreasonable assumption that BRG1 (and many other such factors) engage in opportunistic and largely non-specific association with RNA at the binding sites (see also the latest work by the Guttman lab in Gus et al., Mol Cell 2024). The authors would need to address this in some way, motif analysis being perhaps the easiest.

- The knockdown of 8 lncRNAs to establish proof-of-principle is very welcome. However, there are two aspects that require attention here. First, the authors do not indulge the scenario whereby presence of a lncRNA might act as a local deterrent to BRG1 binding (hence its loss will allow more binding at some sites). Second, they completely fail to make use of their Red-C/ChIP data in their analysis and only stratify based on BRG1 CUT&RUN peaks. This is surprising, as the Red-C data would allow them to interrogate positions that are presumably specifically bound by each of the targeted lncRNAs and assess the core events of depletion (as it is probably unreasonable to assume that the many thousands of changes seen are direct effects of BRG1 mis-binding).

- I am also thinking that, in order to really prove the potential of a given RNA to directly recruit BRG1 to a target site, some additional validation is needed. For instance, the authors could supplement back into lncRNA-knockdown cells different parts of their favorite lncRNA and test how BRG1 recruitment is affected. This does not even have to be done in a genome-wide fashion, but few specific sites could be tested using qPCR.

- The ATAC analysis after PROTAC treatment looks thorough, but suggests that BRG1 and not lncRNA presence at enhancers is critical for accessibility. It would then be nice to see if lncRNA-knockdowns also induce a change in accessibility, which would establish a functional hierarchy for binding.

- The analyses behind Fig. 4 are welcome, yet somewhat incomplete. On the one hand, selective sensitivity of enhancer accessibility to BRG1 PROTACs is nice to see (not completely new), but it is again not linked to any Red-type data. This comes very selectively in Fig. 4, with only a ~15% overlap and again without crossing these to Red-C/ChIP data to confirm binding of the lncRNA in question to these sites. Even if the effective resolution is low, this should be an important test to conduct. If BRG1 binding is lost from an enhancer upon knockdown of a given lncRNA, there should be some evidence that this lncRNA can be mapped to that enhancer. In general, this part of the manuscript is the weakest.

- Finally, one should point out that Hi-C is really not the best approach to map enhancer-promoter interactions, as these are typically underrepresented in Hi-C loop catalogues. The Results section does not contain any mention of how Hi-C data was used to extract these interactions and what their confidence levels are. The Methods explanation is also very brief, somehow implicates the ABC model (unclear to me why this was needed) and also the Hi-C data itself is not described so as to have an idea about their quality and effective resolution. This needs to be remedied.

Reviewer #2

(Remarks to the Author)

Summary: This paper effectively explored the role of RNA-DNA-Protein binding in facilitating localization and function of BRG1, proposing that trans-acting lncRNAs largely contribute to this phenomenon at enhancer sites. The strength of this paper was its use of multiple powerful modalities for assessing chromatin structure and binding of proteins/RNA which included global Red-C, BRG1 RedChIP, BRG1 CUT&RUN, RIP-Seq, and iCLIP. The combination of these modalities helped reduce the amount of non-specificity that typically accompanies investigations into these types of interactions. The manuscript provided convincing evidence that lncRNAs are in fact interacting with BRG1 and the SWI/SNF complex in HUVEC cells, and that perturbation of some specific lncRNAs can affect chromatin accessibility at sites consistent with BRG1 activity. This could lead to better understanding of the SWI/SNF complex and regulation of its targets. However, questions about the overall specificity of the techniques employed in this manuscript remain. Similarly, the generalizability of the proposed mechanism for SWI/SNF targeting should be further explored, especially as it pertains to other cell types.

Comments:

1. Figure 1 panel B shows the amount of unique RNAs detected by the two techniques, Red-C and RedChIP. The figure shows a very similar amount of RNAs (~24k) for each technique. What is the overlap between these two sets of unique RNAs? Based on the description, RedChIP should seemingly be much more specific, and one would expect less overlap with Red-C which looks at global RNA-DNA interactions. If there is a lot of overlap, how do the authors explain this?
2. Along the lines of the comment above, Figure 1 panel C shows unique DNA bins that both techniques display. While there is a smaller set of DNA bins detected in RedChIP vs Red-C, that is still a considerable proportion of overlap if many of those unique bins are shared. Can the authors comment on this? Have they carried out RedChIP with proteins other than BRG1 as a control?
3. Similarly, Figure 1 panel F seems to only show a modest enrichment for BRG1 CUT&RUN peaks in DNA bins between Red-C and RedChIP.
4. Recent studies indicate limitations of CLIP for mapping in vivo RNA-binding sites. A more stringent method, CLAP, has been developed to eliminate non-specific RNA binding (PMID: 38387462). Perhaps the authors should consider using CLAP to further verify their findings.
5. In Figure 2 panel B, the enrichment for lncRNA in the iCLIP coverage is shown after normalizing for RNA-seq expression. The authors may want to adjust for RNA size when analyzing iCLIP enrichment. It would be interesting to see whether there is a trend (larger RNAs are more enriched) and if other RNA types such as the protein coding behave similarly. This could help define whether the SWI/SNF directing processes are lncRNA specific or not.

6. In Figure 2 panel E, the process for identifying high-confidence BRG1 interactions is shown. It would be helpful if the authors provided more details about this process and how any associated thresholds were set.
7. In Figure 2 panel H, the number of peak heights is so tight that the figure is difficult to read. Please improve the appearance for better clarity.
8. From Figure 3 onwards, the use of siRNA was instrumental in drawing conclusions about lncRNA perturbation. Extended Figure 2 confirmed some level of knockdown in all the lncRNAs tested, although the level varied from ~60-90% based on the lncRNA. Given the potential for off-target effects of siRNA, and the potential for this to affect subsequent readouts of chromatin accessibility, have the authors carried out similar experiments using CRISPR-edited cell lines? This would be a cleaner way to assess the knockdown of certain lncRNAs and their functional significance, reducing potential off-target effects and allowing for total removal of lncRNA.
9. In Figure 4, a BRG1/BRM-specific PROTAC degrader was used to verify the role of lncRNA in the mSWI/SNF complex. It is unclear which cell model was used for the AU-15330 treatment. Furthermore, Figure 4 panel D shows that “15.7% of the enhancer losses in response to AU-15330 PROTAC could be attributed to the knockdown of any of the seven lncRNAs.” It is critical to compare not only the total overlap but also the magnitude of changes mediated by lncRNA knockdown and AU-15330. Additionally, the impact of knocking down lncRNA00607 on BRG1 chromatin binding in Figure 4 panel H is not appreciable, and the effect of AU-15330 on ATAC-seq signal is minor, particularly in the highlighted regions.
10. A major question regarding the findings in this paper pertains to the reproducibility of RNA perturbation effects within the same cell type. Can the differentially expressed genes be altered in a consistent manner across experiments in the same cell type? For instance, if knockdown of lncRNA X is believed to reduce expression of gene Y based on nearby enhancer effects, can this change be reproduced in a repeat experiment? Showing this level of consistency would help bolster the findings significantly.
11. Following the comment above, how do the findings translate to different cell types? For two cell types with similar levels of expression of a particular lncRNA, does perturbation of that transcript elicit similar changes to chromatin accessibility and/or downstream gene expression? This question of translatability across cell types was touched upon in Figure 1 panel J, which indicated that differences likely exist. However, if there is overlap, a level of predictability in terms of lncRNA perturbation and its effects in different cell types would bolster the paper’s findings of a potential mechanism for SWI/SNF localization.

Reviewer #3

(Remarks to the Author)

This is an important paper, which I am surprised was not considered for publication in *Nature*, perhaps because its wider implications were not canvassed (see last paragraph below). The study shows that SWI/SNF genomic targeting, mainly at enhancers, is mediated by lncRNAs, providing much needed mechanistic flesh to the growing realization that enhancers express trans-acting RNAs that bind and target protein complexes to specific genomic locations during the trillions of cell fate decisions that are made during development.

The authors demonstrate the specific targeting of the SWI/SNF complexes (focusing on its core subunit BRG1) by lncRNAs to specific genomic locations by different methods, thereby abrogating concerns about false positive interactions. First, they use a combination of Red-C (RNA ends on DNA capture) with chromatin immunoprecipitation (RedChIP) to show that there is genome-wide enrichment of RNA-DNA interactions associated with the SWI/SNF complex. Second, they use CUT&RUN to show a general enrichment of Red-C RNA-DNA interactions at BRG1 binding sites and that BRG1 DNA binding sites are predominantly associated with trans-acting RNAs at enhancers. Third, they use RIP-seq and iCLIP to show that lncRNAs are enriched at BRG1 binding sites.

The role of lncRNAs in SWI/SNF (BRG1) targeting was then confirmed by siRNA knockdowns of 8 lncRNAs (including the well-characterized lncRNAs PVT1, LINC00607 and JPX), which altered the intensity and distribution of BRG1 binding peaks. Moreover, degradation of BRG1 by PROTAC is shown to result in the genome wide loss of open chromatin regions associated with enhancers, and Hi-C analysis shows that PROTAC-sensitive sites are linked to genes that are differentially expressed after lncRNA depletion and PROTAC treatment.

While one might nit-pick individual elements of the study, the methods are well detailed, and the combination makes a compelling case for the conclusion that lncRNAs target SWI/SNF to enhancers, thereby adding important texture to the role of lncRNAs in enhancer action.

The paper is compact, well-written and lucid. My only criticism is that the paper should make mention (in the Discussion) of the well-established fact that enhancers express lncRNAs in the cells in which they are active, that enhancer-derived lncRNAs have been shown to be required for enhancer function, and that the results presented in this paper add to the evidence that lncRNAs are the mediators of enhancer target specificity. While making the worthwhile points that that “*Knowledge of the lncRNAs responsible for SWI/SNF recruitment ... offers a tremendous potential for more tailored, mutation-specific therapeutics ... [and] ... advocates lncRNAs as specific targets for individual cancer therapy*”, the paper would be improved by a stronger finish that considers the implications of its findings for the understanding of the mechanism of action of the genes called enhancers. The latter might also be flagged at the end of the Summary paragraph (“... and add to the evidence that enhancer action is mediated by lncRNAs”; or words to that effect).

Minor points:

1. The references need to be formatted in *Nature Communications* style and checked for completeness. In particular:

2. Reference 6 (Brahma & Henikoff) needs to be updated: *Nature Genetics* **56**, 100–111 (2024).

3. Details need to be provided for reference 15 (Golov et al., 2023): <https://doi.org/10.7554/eLife.91596.1> and reference 18 (Saha et al, 2022): <https://doi.org/10.1101/2022.09.06.506785>.

4. The article number needs to be supplied in Reference 52: *Gigascience* **10**, giab008 (2021). Similarly in reference 71: *eLife* **6**, e21856 (2017) and reference 74: *Bioinformatics* **39**, btad062 (2023).

Version 1:

Reviewer comments:

Reviewer #1

(Remarks to the Author)

The authors have done a very thorough job in addressing my comments, and I thank them for this. The manuscript, in its current form, is mature enough for publication.

Reviewer #2

(Remarks to the Author)

Thank you for incorporating my previous suggestion to include CRISPR knockout (KO) of the lincRNA in addition to the siRNA knockdown. However, the manuscript lacks a description of how the CRISPR KO was performed. Providing this information is essential for a proper interpretation of the data presented in the revised manuscript.

Reviewer #3

(Remarks to the Author)

The revised manuscript adequately addresses my suggestions, and the more detailed comments of the other reviewers.

This is an important paper, both in and of itself, and for its broader implications for understanding the role(s) of lincRNAs in mediating enhancer action.

Version 2:

Reviewer comments:

Reviewer #2

(Remarks to the Author)

Summary: This paper effectively explored the role of RNA-DNA-Protein binding in facilitating localization and function of BRG1, proposing that trans-acting lincRNAs largely contribute to this phenomenon at enhancer sites. The strength of this paper was its use of multiple powerful modalities for assessing chromatin structure and binding of proteins/RNA which included global Red-C, BRG1 RedChIP, BRG1 CUT&RUN, RIP-Seq, and iCLIP. The combination of these modalities helped reduce the amount of non-specificity that typically accompanies investigations into these types of interactions. The manuscript provided convincing evidence that lincRNAs are in fact interacting with BRG1 and the SWI/SNF complex in HUVEC cells, and that perturbation of some specific lincRNAs can affect chromatin accessibility at sites consistent with BRG1 activity. This could lead to better understanding of the SWI/SNF complex and how to control its targeting.

Comments:

All previous concerns have been addressed.

Reply to reviewer 1:

Reviewer 1: *The manuscript by Oo and colleagues investigates the concept of RNA mediating the binding of SWI/SNF complexes to its genomic targets. To address this, the authors perform a large number of NGS experiments to map RNA-DNA, protein-DNA and protein-RNA interactions focusing on the key BRG1 subunit of SWI/SNF complexes. The data is, for the most part, presented clearly and the analyses and figures are informative. The majority of conclusions are carefully drawn, and the implications of these findings come to build on the emerging concept of RNA-guided and/or RNA-assisted binding of TFs to chromatin. I find this contribution to be potentially important and of interest to a broad readership like that of Nat Communications. At the same time, I am listing below a number of points that, in my opinion, need attention in order to either strengthen or further clarify the findings of the manuscript. I hope that the authors find them useful while revising their work.*

Reply: We thank the reviewer for the time and effort spent in reviewing our work and for their positive assessment of the manuscript. We have addressed all comments and have revised the manuscript accordingly, as outlined in the point-by-point reply below.

Reviewer 1 - 1: *The argument in the first paragraph that RedChIP produces more enrichment than Red-C, therefore "demonstrat[ing] a genome-wide enrichment of RNA-DNA interactions with the SWI/SNF complex" is partially flawed. Any immune-selecting method produces by definition a different structure of library for sequencing and, also by definition, a library enriched for the target of the antibody (assuming the antibody is specific and the target complex accessible, which seems to be the case for BRG1). Therefore, this line of argumentation is not correct and the fold-change calculated in Fig. 1d should not be interpreted in such a way. I would argue that RedChIP for BRG1 shows local enrichment for RNA-DNA contacts at BRG1-bound sites and nothing more in comparison to genome-wide Red-C.*

Reply: We appreciate the reviewer's comment that RedChIP inherently enriches for RNA-DNA contacts at BRG1-bound sites due to the specificity of the antibody. However, the statement in the original manuscript refers to genome-wide enrichment in the sense of an increased number of RNA-DNA interactions across multiple BRG1-bound loci, rather than a global increase in interactions across the entire genome. We aimed to highlight the strong overlap between Red-C RNA-DNA sites and BRG1 CUT&RUN peaks even before BRG1 RedChIP was introduced in the manuscript. Red-C, while mapping global RNA-DNA interactions, lacks the specificity of RedChIP for BRG1, which is why we observed differences in the resolution and interaction frequencies between the two methods.

We have since improved the clarity of the description in the manuscript (lines 109–124). To better illustrate this aspect, we have also provided an updated figure (**Extended Figure 1b and Reply 1, figure 1**), showing that an enrichment in RNA binding at BRG1-bound genomic regions can be observed in Red-C, which is absent when the BRG1 CUT&RUN peaks are randomly shuffled throughout the genome.

Reply 1, figure 1. Mean DNA bin interaction frequency in Red-C and BRG1 RedChIP data at regions which overlap with a BRG1 CUT&RUN peak or not, or overlap with a randomly shuffled BRG1 CUT&RUN peak or not.

Reviewer 1 - 2: It would be also informative if the main Fig. 1 included zoomed-in snapshots of Red-C/RedChIP signal distributions from the genome browser, as well as average plots along BRG1 peaks and control non-bound regions. The statement on "limited resolution" of the Red methods would also require an explanation of which resolution is actually being used here and how this was chosen. This is important for interpreting most panels in Fig. 1 where comparisons are made.

Reply: Average plots have now been supplied to demonstrate an enrichment of Red-C and RedChIP RNA-DNA sites at BRG1 CUT&RUN peaks compared to shuffled peaks (**Extended figure 1 and Reply 1, figure 2a**). Regarding the "limited resolution" of the Red methods, we acknowledge the need for further clarification. In this context, "limited resolution" refers to both the expansive genomic search space, and the difficulty in defining an exact resolution at which RNA and DNA ligate. Binning the genome enables more robust comparisons between samples e.g. Red-C vs. RedChIP, whilst maintaining a resolution sufficient for mapping RNA binding to genomic features e.g. enhancers and promoters. 5 kb was the maximum resolution at which we were confident in reliably analyzing the data. While a higher resolution would be ideal, current methodological limitations prevent us from achieving this level of detail genome-wide. We have now clarified this aspect in the text (lines 118-119) and updated **Figure 1k and Reply 1, figure 2b** with expanded browser traces demonstrating the overlap between BRG1 RedChIP and CUT&RUN traces.

Reply 1, figure 2. a, Genome-wide density plots for RNA-DNA interactions for both Red-C and RedChIP DNA coverage at BRG1 CUT&RUN peaks versus a size-matched set of shuffled peaks. b, expanded browser traces demonstrating an overlap between BRG1 RedChIP DNA bins and BRG1 CUT&RUN coverage.

Reviewer 1 - 3: Regarding the composition of RNA-bound positions (presumably at low resolution bins, judging from the scale in Fig. 1k), how many different RNAs can be found bound to any given bin? How does this variety change genome-wide and according to the genomic character (genic v non-genic, HMM composition, histone mark, transcriptional activity, etc)? Do the authors believe that RNA combinations are responsible for inducing binding or rather single RNAs?

Reply: We appreciate the reviewer's question regarding the diversity of RNAs bound to specific genomic regions. This is indeed a critical point but challenging to address with the current data and methodologies. In our data, we detect between 0 and 86 individual RNAs within the 5 kb genomic bins. However, given that these experiments are performed using bulk populations of cells, it is not possible with the current technology to determine whether these RNAs are binding simultaneously at a given locus, or whether different RNAs bind a locus in different cells, introducing a level of redundancy.

Nevertheless, our data show that the majority of genomic bins (~ 55-60%) are bound by a single RNA, with approximately 25% of bins able to bind two different RNAs. This pattern is consistent between Red-C and RedChIP experiments. When considering different chromatin states, we observe that bins intersecting with enhancer regions – as annotated by ChromHMM – tend to be bound more frequently by several different RNAs (more instances of 3+ RNAs detected at the bin). However, we cannot determine from these data whether this combinatorial RNA binding is functionally relevant, or whether it simply reflects diverse RNA binding across different cells. Given recent developments in the study of RNA-RNA interactions and the possibility that RNAs and proteins may form collective scaffolds on chromatin, we believe this could be an important area for future investigation. Unfortunately, our current assays do not offer single-cell resolution, which would be necessary to definitively address the role of combinatorial RNA localization.

Reviewer 1 - 4: The statement that "BRG1 DNA binding sites are predominantly associated with trans-acting RNAs" is somewhat misleading, as the numbers in Fig. 1g actually show a <18% difference in total numbers, with >3500 sites having both cis and trans RNAs mapped to them. And this also brings about another question: at these shared sites, do these cis and trans RNAs coexist? If so, how do the authors imagine them working? Given the strong enrichments of trans RNAs in intergenic regions and the enrichment of cis RNAs at intronic/intragenic regions, which positions have the most trans/cis RNA co-association? Is there any experiment that can be done to dissect this? For example, silencing of the underlying gene with dCas9-KRAB and assessment of how loss of the cis (but perhaps not of the trans RNA) affects BRG1 binding locally?

Reply: We thank the reviewer for raising this important point. Similar to the previous question, it is difficult to definitively conclude whether *cis* and *trans* RNAs are binding simultaneously to the same DNA site or if these "mixed" sites are capable of binding multiple RNAs individually, as suggested in our bulk cell experiments. The heterogeneity of the cell population used in these assays prevents us from determining whether these RNAs are acting in a combinatorial manner at single loci. To resolve this issue, RNA-DNA ligation experiments optimized for single-cell resolution would be required. Unfortunately, this technology is not currently available for Red-C or RedChIP experiments, limiting our ability to make concrete assertions about the coexistence of *cis* and *trans* RNAs at specific loci.

We do observe that mixed *cis/trans* RNA binding sites are most enriched at promoters and 5' UTRs. This likely reflects the local transcription of the *cis* RNA, along with the binding of a *trans* RNA. From the data generated here, we observe examples where the depletion of a *trans* RNA correlates with the loss of BRG1 binding at mixed sites. Two such examples are *LINC00607* & *PVT1*, where the loss of these as *trans* RNAs leads to reduced BRG1 binding at mixed *cis/trans* sites (**Extended Fig. 2b and Reply 1, figure 3**).

In response to the reviewer's suggestion, we also developed a dCas9-KRAB approach for *LINC00607* which binds in *cis* at a mixed site enhancer approximately 1 Kb upstream of the *LINC00607* transcription start site. CUT&RUN revealed a modest decrease in BRG1 binding at this particular site after *LINC00607* CRISPRi (**Reply 1, figure 4**). In comparison, the complete knockout of *LINC00607* lead to a stronger decrease in BRG1 binding at this site but did not completely abolish binding. This may suggest that other RNAs at combinatorial binding sites could be responsible for the remaining BRG1 binding, or again that in a mixed cell population we cannot dissect every RNA that binds at a given genomic locus.

Reply 1, figure 3. Browser traces demonstrating BRG1 CUT&RUN peaks at mixed *cis/trans* RNA binding sites. Depletion of the *trans* RNA e.g. *LINC00607* (top panel) and *PVT1* (bottom panel) reduces BRG1 binding at the genomic locus of the *cis* RNA (*SMG7* and *PCDH9* respectively). Graphs on the right demonstrate the degree of binding for each RNA at the respective mixed *cis/trans* binding site.

Reply 1, figure 4. Browser traces demonstrating BRG1 CUT&RUN peaks at mixed *cis/trans* RNA binding sites. Depletion of the *cis* RNA *LINC00607* with CRISPRi slightly reduces BRG1 binding at the mixed *cis/trans* site. CRISPR KO of *LINC00607* has a stronger effect on BRG1 binding at this mixed site.

Reviewer 1 - 5: Given that RIP-seq is a rather noisy experiment, and that the authors have already performed iCLIP, I think that all mention of RIP can be removed from text and figures as superfluous and of lower confidence. The scheme in Fig. 2e does not justify its use.

Reply: We agree with the reviewer and have removed all mention of RIP-seq from the text and figures.

Reviewer 1 - 6: The iCLIP profiles provided in Fig. 2h are rather broad and look a bit unspecific. There are three things to be clarified here. First, that there is little convergence between RIP and iCLIP (which is another reason why RIP should be removed, in my opinion). Second, the peak calling for PVT1 looks rather unspecific compared to that for LINC00607. Third, what is the motif consensus outcome after using iCLIP peaks to deduce specificity in BRG1 binding? I think it not an unreasonable assumption that BRG1 (and many other such factors) engage in opportunistic and largely non-specific association with RNA at the binding sites (see also the latest work by the Guttman lab in Gus et al., Mol Cell 2024). The authors would need to address this in some way, motif analysis being perhaps the easiest.

Reply: We completely agree with the reviewer that the specificity of RNA-protein interactions is an important aspect.

A lack of complete convergence between the RIP-seq and iCLIP was expected given the different effective resolutions of the methods. Nevertheless, as suggested, we have removed the RIP-seq traces and replaced them with clearer, strand-specific iCLIP tracks for *LINC00607* and *PVT1* (**Fig. 2h and Reply 1, figure 5**).

Reply 1, figure 5. Expanded browser traces demonstrating BRG1 binding sites on *LINC00607* and *PVT1*.

Regarding the apparent lower specificity of PVT1 peaks compared to LINC00607, we believe this is likely due to the much higher relative expression of PVT1, resulting in more BRG1-PVT1 binding events, potentially across more sites. All iCLIP experiments were analyzed using the same peak-calling approach (PureCLIP and BindingSiteFinder) with four biological replicates, so we are confident in the specificity of the identified binding sites.

The reviewer raises an important point on wider protein-RNA binding specificity, a point not limited to BRG1 but relevant to the field in general. To explore potential RNA-binding specificity, we compared the motifs enriched at BRG1 iCLIP binding sites with motifs from a published TDP-43 iCLIP in HUVEC (Hipke et al., 2023, Front. Cell Dev. Biol.) (**Reply 1, figure 6**). As expected, polyA and polyU tracts were detected in both datasets, which are known motifs often resulting from UV-C crosslinking in iCLIP experiments. However, distinct motifs were also identified in regions flanking the binding sites, suggesting unique binding preferences for each protein. For BRG1, motifs such as GUCACUCG, CGAUUAAAA and ACU*GG*UCU were specifically enriched upstream of binding sites, suggesting that BRG1 does exhibit RNA-binding specificity.

Reply 1, figure 6. Motif enrichment analysis for BRG1 iCLIP (top) and TDP-43 (bottom). Both iCLIPs were performed in HUVEC.

We agree with the reviewer that many RNA-protein interactions may be opportunistic or even artefactual. We recently discussed this aspect with the Guttman lab and explored the feasibility and requirement to perform CLAP. Importantly, for this manuscript we could observe a functional consequence of BRG1 genomic binding after depletion of those RNAs found to interact with BRG1. We considered performing CLAP to strengthen the specificity of RNA-protein interaction data but ultimately decided against it for the current study. This decision was mainly due to the complexity of the SWI/SNF complex, with 12+ subunits, and the technical difficulty of individually tagging and purifying each subunit while maintaining the integrity of the complex. Additionally, a recent reanalysis of the CLAP approach by Jeannie Lee's lab (Lee *et al.* 2024, bioRxiv) affirms PRC2 as an RNA binding protein and *XIST* RNA as a target and also questions the added value of CLAP over CLIP, suggesting that CLAP does not provide significantly more specific RNA binding sites than iCLIP.

Nonetheless, after our consultation with the Guttman lab, we performed experiments that combine a BRG1 RIP-qPCR with the highly stringent CLAP washing steps (5 different stringent buffers, 3 washes each at 90°C for 3 minutes). We also performed this for PTBP1 and checked for *CFL1* RNA, an interaction which served as a positive control in the CLAP paper (Guo *et al.* 2024, Mol. Cell). Firstly, *CFL1* was retrieved with PTBP1 under the CLAP conditions in our lab. Additionally, BRG1 RIP with stringent CLAP wash conditions pulled down our key lncRNAs such as *MIR100HG*, *PVT1*, and *LINC00607*, all showing higher recovery rates than *CFL1*. In contrast, we failed to detect RNAs such as *JPX*, *MIR31HG*, and *NEAT1*, suggesting that these RNAs may interact with other SWI/SNF subunits rather than BRG1 (**Reply 1, figure 7**).

Reply 1, figure 7. PTBP1 and BRG1 RIP-qPCR with CLAP washing steps. PTBP1 pull-down of *CFL1* served as a positive control. BRG1-bound lncRNAs *MIR100HG*, *PVT1* and *LINC00607* were amplified.

Reviewer 1 - 7: *The knockdown of 8 lncRNAs to establish proof-of-principle is very welcome. However, there are two aspects that require attention here. First, the authors do not indulge the scenario whereby presence of a lncRNA might act as a local deterrent to BRG1 binding (hence its loss will allow more binding at some sites). Second, they completely fail to make use of their Red-C/ChIP data in their analysis and only stratify based on BRG1 CUT&RUN peaks. This is surprising, as the Red-C data would allow them to interrogate positions that are presumably specifically bound by each of the targeted lncRNAs and assess the core events of depletion (as it is probably unreasonable to assume that the many thousands of changes seen are direct effects of BRG1 mis-binding).*

Reply: We agree that lncRNA deterrence of BRG1 is an important scenario that should be considered. While our primary focus was on BRG1 loss-of-binding events following lncRNA knockdown, we did observe several thousand gained BRG1 binding sites after knockdown, which suggests that certain RNAs might indeed act as local deterrents to BRG1 binding. We have now expanded the discussion of this possibility in the manuscript to acknowledge that RNAs may have dual roles—both promoting and inhibiting BRG1 binding depending on the genomic context (lines 182-183).

The reviewer raises a valid point about leveraging the Red-C/RedChIP data in conjunction with the BRG1 CUT&RUN data. This was investigated further and a modest overlap between RNA-binding sites identified by Red-C/RedChIP and the regions where BRG1 binding changes following lncRNA knockdown was observed (**Reply 1, figure 8**). This overlap, while present, is limited, which we attribute to the relatively shallow sequencing depth of the Red-C/RedChIP experiments. As these methods are designed to capture global RNA-DNA interactions, the sequencing depth is not sufficient to comprehensively map specific RNA interactions at each individual locus, particularly for lower-expressed RNAs. Nevertheless, we agree with the reviewer that it is unlikely that all of the thousands of BRG1 binding changes seen after knockdown are direct effects of lncRNA depletion.

Reply 1, figure 8. Overlap between enhancer RNA-binding sites identified by Red-C/RedChIP and the regions where BRG1 binding changes following lncRNA knockdown.

Reviewer 1 - 8: I am also thinking that, in order to really prove the potential of a given RNA to directly recruit BRG1 to a target site, some additional validation is needed. For instance, the authors could supplement back into IncRNA-knockdown cells different parts of their favorite IncRNA and test how BRG1 recruitment is affected. This does not even have to be done in a genome-wide fashion, but few specific sites could be tested using qPCR.

Reply: We appreciate the reviewer's suggestion, and agree that rescue experiments are useful to further validate the role of IncRNAs in recruiting BRG1 to specific genomic sites.

To investigate the functional domains of the IncRNAs responsible for BRG1 recruitment, we designed plasmids with various deletion mutants of the IncRNAs. After knockdown, these mutants were overexpressed, and we performed BRG1 CUT&RUN. However, the resulting CUT&RUN traces from the deletion mutants showed an intermediate pattern between the native and knockdown situation, rather than revealing individual segments responsible for interactions. This makes it difficult to draw definitive conclusions regarding specific RNA fragments' ability to rescue BRG1 binding. We believe that dissecting the exact domains responsible for RNA-BRG1 interactions will require more detailed follow-up experiments. From the iCLIP data, we know that each IncRNA contains multiple BRG1 binding sites, potentially with regions involved in DNA binding, RNA modifications, and other protein interactions all within a complex tertiary RNA structure. This is compatible with the results observed with the deletion mutants. Deleting specific stretches of the transcript will also have unintended consequences on RNA folding or other aspects of its function, which we cannot fully control for at this stage.

Nevertheless, we also agree with the reviewer that a demonstration of a specific function of the IncRNA in a rescue experiment is needed. Therefore, plasmids containing either full-length *JPX* or a *MIR100HG* transcript were generated and overexpressed in HUVECs following siRNA-mediated knockdown of the respective IncRNA. Instead of using qPCR at selected sites, we opted to perform BRG1 CUT&RUN experiments with sequencing to reveal a bigger picture of the rescue of BRG1 binding. Remarkably, overexpression of *JPX* and *MIR100HG* did indeed rescue BRG1 binding at many genomic sites (**Fig. 3g and Reply 1, figure 9**).

Reply 1, figure 9. Browser traces of BRG1 CUT&RUN experiments after siRNA-mediated knockdown of *MIR100HG* (up) or *JPX* (down) and overexpression (rescue) with pcDNA3.1+*MIR100HG* (up) or pcDNA3.1+*JPX* (down). For each, five individual genomic loci are shown.

Reviewer 1 - 9: The ATAC analysis after PROTAC treatment looks thorough, but suggests that BRG1 and not lncRNA presence at enhancers is critical for accessibility. It would then be nice to see if lincRNA-knockdowns also induce a change in accessibility, which would establish a functional hierarchy for binding.

Reply: We thank the Reviewer for raising this excellent point. To address this, we performed ATAC-seq after knockdown of the seate BRG1-bound lncRNA *MIR100HG*. Upon differential peak analysis, we observed that 67% of 111 ATAC peaks downregulated after *MIR100HG* knockdown corresponded to peaks which were also lost following treatment of HUVECs with a BRG1-targeting PROTAC. This suggests that at a subset of genomic loci, the presence of the lncRNA is important for maintaining chromatin accessibility through BRG1 recruitment. However, based on the current data, we cannot make a definitive statement about the functional hierarchy of lncRNAs and BRG1 in chromatin accessibility.

We also provide examples of where these PROTAC- and si*MIR100HG*-sensitive peaks overlap with enhancers, as classified by ChromHMM (**Extended Fig. 3b and Reply 1, figure 10**). Furthermore, we compared our previously published ATAC-seq data from CRISPR-mediated knockout of *LINC00607* (Boos et al. 2023, BRIC) to the ATAC-seq after BRG1 PROTAC treatment. Again, we observed several enhancer regions that were sensitive to both *LINC00607* knockout and BRG1 depletion.

Reply 1, figure 10. Browser traces of ATAC-seq in HUVEC after AU-15330 PROTAC treatment, siRNA-mediated *MIR100HG* knockdown or LentiCRISPRv2-mediated *LINC00607* KO showing three individual genomic loci. NTC, non-targeting control gRNA.

Reviewer 1 - 10: The analyses behind Fig. 4 are welcome, yet somewhat incomplete. On the one hand, selective sensitivity of enhancer accessibility to BRG1 PROTACs is nice to see (not completely new), but it is again not linked to any Red-type data. This comes very selectively in Fig. 4, with only a ~15% overlap and again without crossing these to Red-C/ChIP data to confirm binding of the lncRNA in question to these sites. Even if the effective resolution is low, this should be an important test to conduct. If BRG1 binding is lost from an enhancer upon knockdown of a given lncRNA, there should be some evidence that this lncRNA can be mapped to that enhancer. In general, this part of the manuscript is the weakest.

Reply: We agree with the Reviewer that linking the enhancer accessibility data to Red-C/RedChIP data is important. As the Reviewer points out, the relatively low effective resolution and sequencing depth of

the Red-C/RedChIP methods contribute to the limited overlap (~15%) between these datasets and enhancer accessibility data.

Despite that, *MIR100HG* was found to bind in proximity to enhancers that were sensitive to both BRG1 PROTAC treatment and to *MIR100HG* knockdown (**Reply 1, figure 11**).

Reply 1, figure 11. Browser traces of ATAC-seq in HUVEC after AU-15330 PROTAC treatment or siRNA-mediated *MIR100HG* knockdown at a BRG1-*MIR100HG* RedChIP binding site. Chromatin compaction can be seen at enhancers sensitive to both PROTAC and siMIR100HG.

Reviewer 1 - 11: Finally, one should point out that Hi-C is really not the best approach to map enhancer-promoter interactions, as these are typically underrepresented in Hi-C loop catalogues. The Results section does not contain any mention of how Hi-C data was used to extract these interactions and what their confidence levels are. The Methods explanation is also very brief, somehow implicates the ABC model (unclear to me why this was needed) and also the Hi-C data itself is not described so as to have an idea about their quality and effective resolution. This needs to be remedied.

Reply: We understand the reviewer's concerns. It is true that Hi-C may underrepresent short-range enhancer-promoter loops compared to specialized methods like HiChIP or Capture-Hi-C. Hi-C, however, provides a more comprehensive view of the 3D genome, capturing all chromatin interactions without being restricted to pre-selected target regions (Rao *et al.*, 2014; Dixon *et al.*, 2015). Capture-based methods can be advantageous when sequencing depth is a limiting factor, but they lack the unbiased coverage that Hi-C provides. We also have HiChIP established in our lab (Warwick *et al.* NAR, 2022), but considering this specific aspect, we actively decided for Hi-C in the present study.

To predict functional enhancer-promoter contacts, we indeed applied the ABC (Activity-By-Contact) model (Fulco *et al.*, 2019), which integrates gene expression, ChIP-seq/CUT&RUN and Hi-C data to assess potential enhancer activity. This model ranks enhancers based on their chromatin accessibility, histone modifications (such as H3K27ac), and proximity to promoters via Hi-C reads, providing a robust framework for identifying enhancer-promoter interactions. The ABC model has been widely validated and used in multiple high-impact studies (Fulco *et al.*, 2019; Nasser *et al.*, 2021), making it a reliable approach for predicting functional interactions from our data.

We acknowledge that the description of our Hi-C data and its analysis in the manuscript should be expanded. We have therefore provided a more detailed explanation of how the ABC model was applied in this context, to ensure the methodology is clear and transparent (lines 579-582).

Reply to reviewer 2:

Reviewer 2: Summary: This paper effectively explored the role of RNA-DNA-Protein binding in facilitating localization and function of BRG1, proposing that trans-acting lncRNAs largely contribute to this phenomenon at enhancer sites. The strength of this paper was its use of multiple powerful modalities for assessing chromatin structure and binding of proteins/RNA which included global Red-C, BRG1 RedChIP, BRG1 CUT&RUN, RIP-Seq, and iCLIP. The combination of these modalities helped reduce the amount of non-specificity that typically accompanies investigations into these types of interactions.

The manuscript provided convincing evidence that lncRNAs are in fact interacting with BRG1 and the SWI/SNF complex in HUVEC cells, and that perturbation of some specific lncRNAs can affect chromatin accessibility at sites consistent with BRG1 activity. This could lead to better understanding of the SWI/SNF complex and regulation of its targets. However, questions about the overall specificity of the techniques employed in this manuscript remain. Similarly, the generalizability of the proposed mechanism for SWI/SNF targeting should be further explored, especially as it pertains to other cell types.

Reply: We appreciate the reviewer's positive feedback. We have addressed the comments and have revised the manuscript accordingly, as outlined in a point-by-point-reply below.

Reviewer 2 - 1: Figure 1 panel B shows the amount of unique RNAs detected by the two techniques, Red-C and RedChIP. The figure shows a very similar amount of RNAs (~24k) for each technique. What is the overlap between these two sets of unique RNAs? Based on the description, RedChIP should seemingly be much more specific, and one would expect less overlap with Red-C which looks at global RNA-DNA interactions. If there is a lot of overlap, how do the authors explain this?

Reply: This is an important point raised by the reviewer. As noted, we do observe similar numbers of unique RNAs (~24,000) detected by both Red-C and BRG1 RedChIP. There is also a significant overlap between the RNA populations identified in these two assays, which can be attributed to the shared RNA repertoire present in the cell for both techniques. However, this overlap does not take into account the relative abundance of each RNA in each assay. When we consider the total frequency of each RNA, we observe that Red-C yields more unique RNAs of lower frequency, while RedChIP is enriched for more RNAs with a higher frequency.

Given the widespread binding of BRG1 across the genome, including at regulatory regions and sites of active transcription, it is not unexpected that RedChIP may pull down background RNAs in addition to BRG1-specific interactions. This background likely contributes to the observed overlap. However, when we focus on RNAs with higher summed frequencies (i.e., those RNAs that are more frequently associated with BRG1-bound regions), the specificity of BRG1 RedChIP becomes clearer (**Extended Fig. 1d and Reply 2, figure 1**).

Reply 2, figure 1. Venn diagram demonstrating the overlap of RNAs between Red-C and BRG1 RedChIP and the number of RNAs for multiple ranges of summed RNA frequency between Red-C and BRG1 RedChIP. More RNAs are identified with BRG1 RedChIP than Red-C as the summed RNA frequency increases.

Reviewer 2 - 2: Along the lines of the comment above, Figure 1 panel C shows unique DNA bins that both techniques display. While there is a smaller set of DNA bins detected in RedChIP vs Red-C, that is still a considerable proportion of overlap if many of those unique bins are shared. Can the authors comment on this? Have they carried out RedChIP with proteins other than BRG1 as a control?

Reply: We appreciate the reviewer's comment and agree that there is a considerable proportion of DNA bins shared between Red-C and BRG1 RedChIP. However, the overlap is not as extensive as that observed with RNA (discussed above). This is expected, as BRG1 RedChIP identifies a more specific subset of DNA regions enriched for BRG1 binding, while Red-C captures global RNA-DNA interactions.

Some commonality between the two datasets is anticipated, given that BRG1 RedChIP is expected to return an enriched subset of the global DNA interaction sites detected in Red-C. When examining the relative frequencies of DNA bins in both assays, we observe that Red-C detects a broader range of bins at lower frequencies, whereas RedChIP identifies a greater number of high-frequency bins (**Extended Fig. 1e and Reply 2, figure 2**). These high-frequency bins are likely enriched for BRG1 binding, reinforcing the idea that BRG1 RedChIP is more specific in detecting BRG1-associated DNA regions.

Reply 2, figure 2. Venn diagram demonstrating the overlap of DNA bins between Red-C and BRG1 RedChIP and the number of DNA bins for multiple ranges of summed DNA frequency between Red-C and BRG1 RedChIP. More DNA bins are identified with BRG1 RedChIP than Red-C as the summed DNA frequency increases.

As suggested by the reviewer, we also conducted a further RedChIP experiment in HUVECs using an alternative protein target, CTCF. When comparing the DNA regions identified in CTCF RedChIP to those in BRG1 RedChIP and Red-C, we observed a notable number of unique DNA regions detected only in the CTCF RedChIP, indicating the specificity of the method for different proteins (**Reply 2, figure 3**). There was also considerable overlap between regions detected in CTCF RedChIP and Red-C, as expected, given that Red-C captures genome-wide interactions. Importantly, we observed much less overlap between the two different RedChIP experiments (BRG1 and CTCF), demonstrating that these assays are capable of identifying protein-specific RNA-DNA interactions and confirming the specificity of RedChIP for different target proteins.

Reply 2, figure 3. Venn diagram demonstrating the overlap of DNA bins between Red-C, BRG1 RedChIP and CTCF RedChIP. As with Red-C and BRG1 RedChIP, the CTCF RedChIP was also performed in HUVEC and sequenced at 100 mio reads.

Reviewer 2 - 3: Similarly, Figure 1 panel F seems to only show a modest enrichment for BRG1 CUT&RUN peaks in DNA bins between Red-C and RedChIP.

Reply: We appreciate the reviewer's observation regarding the modest enrichment of BRG1 CUT&RUN peaks in DNA bins between Red-C and RedChIP. It is important to note that some overlap between Red-C and BRG1 CUT&RUN peaks is expected, as BRG1 binds at sites of RNA-DNA interactions. Red-C identifies global RNA-DNA interactions, and given BRG1's widespread role in transcriptional regulation, an overlap at those BRG1 sites is not unexpected. However, the RedChIP enriches for RNA-DNA interactions specifically at BRG1-bound loci and we therefore observe further enrichment in comparison to Red-C. Red-C, while mapping global RNA-DNA interactions, lacks the specificity of RedChIP for BRG1, which is why there are still differences in resolution and interaction frequencies observed between the two methods.

This is further supported by the CTCF RedChIP experiment mentioned above, where there was far more overlap with Red-C and also unique interactions specific to CTCF, reinforcing the specificity of RedChIP for different target proteins.

We have also provided a new figure (**Extended Fig. 1b and Reply 2, figure 4**), showing that DNA bin enrichment at BRG1-bound genomic regions can be observed in Red-C. This enrichment is absent when BRG1 CUT&RUN peaks are randomly shuffled throughout the genome.

Reply 2, figure 4. Mean DNA bin interaction frequency in Red-C and BRG1 RedChIP data at regions which overlap with a BRG1 CUT&RUN peak or not, or overlap with a randomly shuffled BRG1 CUT&RUN peak or not.

Reviewer 2 - 4: Recent studies indicate limitations of CLIP for mapping *in vivo* RNA-binding sites. A more stringent method, CLAP, has been developed to eliminate non-specific RNA binding (PMID: 38387462). Perhaps the authors should consider using CLAP to further verify their findings.

Reply: We agree with the reviewer that many RNA-protein interactions may be opportunistic or even artefactual. We recently discussed this aspect with the Guttman lab and explored the feasibility and requirement to perform CLAP. Importantly, for this manuscript we could observe a functional consequence of BRG1 genomic binding after depletion of those RNAs found to interact with BRG1. We considered performing CLAP to strengthen the specificity of RNA-protein interaction data but ultimately decided against it for the current study. This was mainly due to the complexity of the SWI/SNF complex, with 12+ subunits, and the technical difficulty of individually tagging and purifying each subunit while maintaining the integrity of the complex. Additionally, a recent reanalysis of the CLAP approach by Jeannie Lee's lab (Lee *et al.* 2024, bioRxiv) affirms PRC2 as an RNA binding protein and *XIST* RNA as a target and also questions the added value of CLAP over CLIP, suggesting that CLAP does not provide significantly more specific RNA binding sites than iCLIP.

Nonetheless, after our consultation with the Guttman lab, we performed experiments that combine a BRG1 RIP-qPCR with the highly stringent CLAP washing steps (5 different stringent buffers, 3 washes each at 90°C for 3 minutes). We also performed this for PTBP1 and checked for *CFL1* RNA, an interaction which served as a positive control in the CLAP paper (Guo *et al.* 2024, Mol. Cell). We also observed a pulldown of *CFL1* with PTBP1. Additionally, BRG1 RIP under stringent conditions pulled down key lncRNAs such as *MIR100HG*, *PVT1*, and *LINC00607*, all showing a higher % recovery than *CFL1*. In contrast, we failed to detect RNAs such as *JPX*, *MIR31HG*, and *NEAT1*, suggesting that these RNAs may interact with other SWI/SNF subunits rather than BRG1 (**Reply 2, figure 5**).

Reply 2, figure 5. PTBP1 and BRG1 RIP-qPCR with CLAP washing steps. PTBP1 pulldown of *CFL1* served as a positive control. BRG1-bound lncRNAs *MIR100HG*, *PVT1* and *LINC00607* were amplified.

Characterizing the RNA-binding profile of the SWI/SNF complex with greater specificity is important. Even if CLAP is unsuitable we have performed additional experiments in that direction to better characterize the BRG1-RNA binding capacity. We have now purified a His-tagged BRG1 and cloned, *in vitro* transcribed and biotinylated a few of the BRG1-bound lncRNAs (full length and deletion mutants). *in vitro* and semi-*in vitro* binding assays were performed as previously described in our recent publication (Oo *et al.* 2022, Cell Reports). Purified recombinant His-BRG1 was incubated with *in vitro* transcribed biotin-*LINC00607*, biotin-*LINC00607*_Exon7, biotin-*JPX*, biotin-*MIR100HG* or biotin-control RNA (transcribed from the pcDNA3.1+ plasmid) at equimolar concentrations. A streptavidin bead pull-down of the RNAs was performed, followed by immunoblotting of BRG1. For the semi-*in vitro* assay, the biotin-RNAs were incubated in HUVEC lysate and endogenous BRG1 immunoblotted after streptavidin bead pull-down of the RNAs. For both assays, BRG1 could be strongly detected after pull-down of each of the lncRNAs and also with the *LINC00607* Exon 7 transcript (which was originally chosen due to that containing one of the strongest BRG1 iCLIP peaks) (**Reply 2, figure 6**). Only a very faint BRG1 band could be detected with the biotin-control RNA in each assay, demonstrating that even without UV-crosslinking, BRG1 binds the lncRNAs reported in this study.

Reply 2, figure 6. Western blots following immunoprecipitation of biotin-tagged RNA after an *in vitro* binding assay (top) and a semi-*in vitro* binding assay (bottom). For the *in vitro* binding assay, His-tagged BRG1 was incubated with *in vitro* transcribed biotin-*LINC00607* (607), biotin-*LINC00607*_Exon7 (607 Ex 7), biotin-*JPX*, a long transcript of biotin-*MIR100HG* (MIR100) or biotin-control RNA (transcribed from the pcDNA3.1+ plasmid) at equimolar concentrations. As a control, each constituent was also individually incubated. His-tagged BRG1 was detected by staining for BRG1 after pull-down of the biotin-RNA with streptavidin beads. The semi-*in vitro* binding assay was performed in HUVEC nuclear lysate and endogenous BRG1 stained after pull-down of biotin-RNA with streptavidin beads.

To further explore potential RNA-binding specificity, we compared the motifs enriched at BRG1 iCLIP binding sites with motifs from a published TDP-43 iCLIP in HUVEC (Hipke *et al.*, 2023, eCollection) (**Reply 1, figure 7**). As expected, polyA and polyU tracts were detected in both datasets, which are known sequence artifacts from iCLIP experiments. However, distinct motifs were also identified in regions flanking the binding sites, suggesting unique binding preferences for each protein. For BRG1, motifs such as GUCACUCG, CGAUUAAAA and ACU*GG*UCU were specifically enriched upstream of binding sites, suggesting that BRG1 does exhibit RNA-binding specificity.

Reply 2, figure 7. Motif enrichment analysis for BRG1 iCLIP (top) and TDP-43 (bottom). Both iCLIPs were performed in HUVEC.

Reviewer 2 - 5: In Figure 2 panel B, the enrichment for lncRNA in the iCLIP coverage is shown after normalizing for RNA-seq expression. The authors may want to adjust for RNA size when analyzing iCLIP enrichment. It would be interesting to see whether there is a trend (larger RNAs are more enriched) and if other RNA types such as the protein coding behave similarly. This could help define whether the SWI/SNF directing processes are lncRNA specific or not.

Reply: This is a valid point, and we have included this normalization step in our analysis. We found that when iCLIP and RNA coverage were length-normalized, the enrichment in lncRNA-BRG1 association compared to other RNA biotypes was further increased (**Fig. 2 and Reply 2, figure 8**).

Reply 2, figure 8. Correlation between BRG1 iCLIP RPKM and RNA-seq RPKM after RNA length normalization. b. Analysis of RNA classes based on iCLIP RPKM normalized to RNA length and RNA expression. $\log_2(\text{iCLIP RPKM}/\text{RNA-seq RPKM})$ values are plotted to compare RNA classes (lncRNAs, mRNAs, snoRNAs, snRNAs, and miRNAs).

Reviewer 2 - 6: In Figure 2 panel E, the process for identifying high-confidence BRG1 interactions is shown. It would be helpful if the authors provided more details about this process and how any associated thresholds were set.

Reply: We thank the reviewer for this question and apologize for not having made it clearer. We have now updated the manuscript text and **Fig. 2e and f** accordingly. We wanted to approach this in an unbiased manner and therefore selected a handful of lncRNAs that were ranked the highest based on a few criteria. We first filtered for lncRNAs that were present in both Red-C and BRG1 RedChIP datasets, present in the BRG1 iCLIP dataset and expressed across HUVEC replicates in the RNA-seq dataset.

lncRNAs were then ranked compared to a mean scaled value for all lncRNAs for their degree of iCLIP coverage (RPKM), their expression level (RPKM), normalized enrichment score (iCLIP normalized to expression and RNA length) and number of iCLIP binding sites. The 16 lncRNA candidates at the top of this list were higher expressed, enriched and contained more binding sites compared to the mean of all lncRNAs.

Reviewer 2 - 7: In Figure 2 panel H, the number of peak heights is so tight that the figure is difficult to read. Please improve the appearance for better clarity.

Reply: We have removed the RIP-seq traces based on feedback from another reviewer, leaving more space for improved, strand-specific iCLIP traces:

Reply 2, figure 9. Expanded browser traces demonstrating BRG1 binding sites on LINC00607 and PVT1.

Reviewer 2 - 8: From Figure 3 onwards, the use of siRNA was instrumental in drawing conclusions about lncRNA perturbation. Extended Figure 2 confirmed some level of knockdown in all the lncRNAs tested, although the level varied from ~60-90% based on the lncRNA. Given the potential for off-target effects of siRNA, and the potential for this to affect subsequent readouts of chromatin accessibility, have the authors carried out similar experiments using CRISPR-edited cell lines? This would be a cleaner way to assess the knockdown of certain lncRNAs and their functional significance, reducing potential off-target effects and allowing for total removal of lncRNA.

Reply: Yes, we conducted BRG1 CUT&RUN in HUVEC after CRISPR-Cas9-mediated knockout of *LINC00607*. This showed a remarkable similarity of the two methods with respect to the effects on BRG1 peaks, as illustrated by the exemplary traces below (**Reply 2, figure 10**).

Reply 2, figure 10. Browser traces demonstrating BRG1 CUT&RUN traces after CRISPR-mediated knockout of *LINC00607* and siRNA-mediated knockdown of *LINC00607* at exemplary enhancer sites.

Reviewer 2 - 9: In Figure 4, a BRG1/BRM-specific PROTAC degrader was used to verify the role of lncRNA in the mSWI/SNF complex. It is unclear which cell model was used for the AU-15330 treatment. Furthermore, Figure 4 panel D shows that “15.7% of the enhancer losses in response to AU-15330 PROTAC could be attributed to the knockdown of any of the seven lncRNAs.” It is critical to compare not only the total overlap but also the magnitude of changes mediated by lncRNA knockdown and AU-15330. Additionally, the impact of knocking down lncRNA00607 on BRG1 chromatin binding in Figure 4 panel H is not appreciable, and the effect of AU-15330 on ATAC-seq signal is minor, particularly in the highlighted regions.

Reply: The AU-15330 PROTAC was used in HUVEC in order to keep the cell model consistent throughout the manuscript. We thank the reviewer for their point regarding the comparison of the magnitude of changes mediated by lncRNA knockdown versus PROTAC treatment on BRG1-bound enhancers. We have since performed ATAC-seq after knockdown of *MIR100HG* and *JPX* and compared the magnitude (Log_2 Fold Change) of changes with the ATAC-seq for the AU-15330 PROTAC. Despite comparing two very different approaches and not accounting for secondary effects, each siRNA still results in the loss of a subset of BRG1-sensitive enhancers (**Reply 2, figure 11**).

Reply 2, figure 11. Log_2 Fold Change of siMIR100HG/siControl (left) and siJPX/siControl (right) plotted against the log_2 Fold Change of AU-15330 PROTAC/Control ATAC peaks.

Regarding the reviewer’s comment on figure 4H, the traces illustrate that *LINC00607* perturbation only changes BRG1 binding at the enhancer region, while BRG1 binding at the target promoters remains

unchanged. Despite this, the target genes are differentially expressed after *LINC00607* knockdown, supporting the hypothesis that lncRNA depletion and the consequent loss of BRG1 at the enhancer is sufficient to inhibit expression of the target genes. The reviewer is correct regarding the minor effect of the PROTAC on the ATAC peaks at the target gene promoters. The PROTAC efficiently degrades BRG1 at these sites yet the chromatin remains open (only the chromatin at the enhancer region collapses). This is an interesting phenomenon that was recently reported for SWI/SNF target promoters whose accessibility can be reestablished by EP400 following the loss of SWI/SNF (Martin *et al.*, 2023, Cell).

We have provided expanded browser traces for the reviewer's benefit (**Reply 2, figure 12**).

Reply 2, figure 12. Browser traces of ATAC-seq following AU-15330 PROTAC treatment, and BRG1 CUT&RUN after AU-15330 PROTAC treatment or siRNA-mediated knockdown of *LINC00607*. ChromHMM indicates the enhancer in the right-most panel where BRG1 binding is less following PROTAC treatment or *LINC00607* knockdown. Chromatin accessibility at this enhancer is also sensitive to PROTAC treatment. Hi-C indicates the gene promoters to which the lost enhancer targets. These promoters are not sensitive to PROTAC treatment (despite a loss of BRG1). All genes of the corresponding target promoters are differentially expressed following *LINC00607* knockdown.

Reviewer 2 - 10: A major question regarding the findings in this paper pertains to the reproducibility of RNA perturbation effects within the same cell type. Can the differentially expressed genes be altered in a consistent manner across experiments in the same cell type? For instance, if knockdown of *lncRNA X* is believed to reduce expression of gene *Y* based on nearby enhancer effects, can this change be reproduced in a repeat experiment? Showing this level of consistency would help bolster the findings significantly.

Reply: We apologize to the reviewer for not having stressed this aspect sufficiently. The study was performed with a primary cell (human umbilical vein endothelial cells). We used three different batches originating from different donors. As exemplified in **Reply 2, figure 13**, the effects were remarkably similar between the different donors.

Reply 2, figure 13. Separated browser traces for individual HUVEC donors (1, 2 and 3). ATAC-seq following AU-15330 PROTAC treatment, and BRG1 CUT&RUN after AU-15330 PROTAC treatment or siRNA-mediated knockdown of *LINC00607*.

Reviewer 2 - 11: Following the comment above, how do the findings translate to different cell types? For two cell types with similar levels of expression of a particular lncRNA, does perturbation of that transcript elicit similar changes to chromatin accessibility and/or downstream gene expression? This question of translatability across cell types was touched upon in Figure 1 panel J, which indicated that differences likely exist. However, if there is overlap, a level of predictability in terms of lncRNA perturbation and its effects in different cell types would bolster the paper's findings of a potential mechanism for SWI/SNF localization.

Reply: This is an important point raised by the reviewer. We first identified two other cell types with similar expression levels of lncRNAs identified in this study. *MIR100HG* had a ct value of 27 in HUVEC, 27 in human microvascular endothelial cells (HMEC) and 26 in smooth muscle cells (SMC). *JPX* had a ct value of 30 in HUVEC, 31 in HMEC and 30 in SMC.

We then established ATAC-seq in HMEC and SMC following the depletion of *MIR100HG* and *JPX* and compared this to our previous HUVEC ATAC-seq. In general, the sequencing quality wasn't as good as that of HUVECs due to a lower sequencing depth, however, we can show example ATAC-seq traces across the cell types. In the left-most example in **Reply 2, figure 14**, knockdown of *JPX* led to the strongest attenuation in ATAC signal in HUVEC and HMEC. Interestingly, at the enhancer within *ARHGEF11*, both *JPX* and *MIR100HG* depletion led to a strong decrease in ATAC signal but only in HMEC. At both the *C3orf52* and *CPQ* promoters in SMC, the ATAC peaks had slightly different profiles following knockdown of *JPX* and *MIR100HG* respectively. In general we do see loci with similar ATAC peaks between the cell types and loci with differential peaks between the cell types and lncRNA knockdowns.

Reply 2, figure 14. Browser traces of ATAC-seq following siRNA-mediated knockdown of *MIR100HG* and *JPX* in HUVEC, human microvascular endothelial cells (HMEC) and smooth muscle cells (SMC). Five different genomic sites demonstrate differential ATAC-seq peaks depending on the cell type in which the lncRNA was depleted.

Reply to reviewer 3:

Reviewer 3: *This is an important paper, which I am surprised was not considered for publication in Nature, perhaps because its wider implications were not canvassed (see last paragraph below). The study shows that SWI/SNF genomic targeting, mainly at enhancers, is mediated by lncRNAs, providing much needed mechanistic flesh to the growing realization that enhancers express trans-acting RNAs that bind and target protein complexes to specific genomic locations during the trillions of cell fate decisions that are made during development.*

The authors demonstrate the specific targeting of the SWI/SNF complexes (focusing on its core subunit BRG1) by lncRNAs to specific genomic locations by different methods, thereby abrogating concerns about false positive interactions. First, they use a combination of Red-C (RNA ends on DNA capture) with chromatin immunoprecipitation (RedChIP) to show that there is genome-wide enrichment of RNA-DNA interactions associated with the SWI/SNF complex. Second, they use CUT&RUN to show a general enrichment of Red-C RNA-DNA interactions at BRG1 binding sites and that BRG1 DNA binding sites are predominantly associated with trans-acting RNAs at enhancers. Third, they use RIP-seq) and iCLIP to show that lncRNAs are enriched at BRG1 binding sites.

The role of lncRNAs in SWI/SNF (BRG1) targeting was then confirmed by siRNA knockdowns of 8 lncRNAs (including the well-characterized lncRNAs PVT1, LINC00607 and JPX), which altered the intensity and distribution of BRG1 binding peaks. Moreover, degradation of BRG1 by PROTAC is shown to result in the genome wide loss of open chromatin regions associated with enhancers, and Hi-C analysis shows that PROTAC-sensitive sites are linked to genes that are differentially expressed after lncRNA depletion and PROTAC treatment.

While one might nit-pick individual elements of the study, the methods are well detailed, and the combination makes a compelling case for the conclusion that lncRNAs target SWI/SNF to enhancers, thereby adding important texture to the role of lncRNAs in enhancer action.

The paper is compact, well-written and lucid. My only criticism is that the paper should make mention (in the Discussion) of the well-established fact that enhancers express lncRNAs in the cells in which they are active, that enhancer-derived lncRNAs have been shown to be required for enhancer function, and that the results presented in this paper add to the evidence that lncRNAs are the mediators of enhancer target specificity. While making the worthwhile points that that “Knowledge of the lncRNAs responsible for SWI/SNF recruitment ... offers a tremendous potential for more tailored, mutation-specific therapeutics ... [and] ... advocates lncRNAs as specific targets for individual cancer therapy”, the paper would be improved by a stronger finish that considers the implications of its findings for the understanding of the mechanism of action of the genes called enhancers. The latter might also be flagged at the end of the Summary paragraph (“... and add to the evidence that enhancer action is mediated by lncRNAs”, or words to that effect).

Reply: We appreciate the reviewer’s very positive feedback. We have addressed the comments and have revised the manuscript accordingly. The summary paragraph and discussion have been improved after following the reviewer’s recommendations.

Minor points:

Reviewer 3 - 1: *The references need to be formatted in Nature Communications style and checked for completeness. In particular: Reference 6 (Brahma & Henikoff) needs to be updated: Nature Genetics 56, 100–111 (2024).*

Reply: We thank the reviewer for this point and have now formatted the references in the style of Nature Communications. Details for this particular reference have been provided.

Reviewer 3 - 2: *Details need to be provided for reference 15 (Golov et al., 2023): <https://doi.org/10.7554/eLife.91596.1> and reference 18 (Saha et al, 2022): <https://doi.org/10.1101/2022.09.06.506785>.*

Reply: Details for these references have been provided, including for reference 18 which now references the recently published article and not the pre-print.

Reviewer 3 - 3: *The article number needs to be supplied in Reference 52: Gigascience 10, giab008 (2021). Similarly in reference 71: eLife 6, e21856 (2017) and reference 74: Bioinformatics 39, btad062 (2023).*

Reply: We have updated these references with article numbers.

We thank the reviewers for taking the time to review our manuscript and for their positive feedback.

Reply to reviewer 2:

Reviewer 2: Thank you for incorporating my previous suggestion to include CRISPR knockout (KO) of the lincRNA in addition to the siRNA knockdown. However, the manuscript lacks a description of how the CRISPR KO was performed. Providing this information is essential for a proper interpretation of the data presented in the revised manuscript.

Reply: We thank the reviewer again for taking the time to review our manuscript. We have now included more details of the CRISPR/Cas9 approach in the methods section of the manuscript. The same approach was described in one of our previous publications (Boos *et al.*, 2023, BRIC).